# *KDM5A* mutations identified in autism spectrum disorder using forward genetics

**Lauretta El Hayek**[1], **Islam Oguz Tuncay**[2], **Nadine Nijem**[1], **Jamie Russell**[3], **Sara Ludwig**[3], **Kiran Kaur**[1], **Xiaohong Li**[3], **Priscilla Anderton**[3], **Miao Tang**[3], **Amanda Gerard**[4,5], **Anja Heinze**[6], **Pia Zacher**[6,7], **Hessa S Alsaif**[8], **Aboulfazl Rad**[9], **Kazem Hassanpour**[10], **Mohammad Reza Abbaszadegan**[11,12], **Camerun Washington**[13], **Barbara R DuPont**[13], **Raymond J Louie**[13], **CAUSES Study**[14], **Madeline Couse**[14], **Maha Faden**[15], **R Curtis Rogers**[13], **Rami Abou Jamra**[6], **Ellen R Elias**[16], **Reza Maroofian**[17], **Henry Houlden**[17], **Anna Lehman**[14], **Bruce Beutler**[3], **Maria H Chahrour**[1,2,3,18,19]*

[1]Eugene McDermott Center for Human Growth and Development, University of Texas Southwestern Medical Center, Dallas, United States; [2]Department of Neuroscience, University of Texas Southwestern Medical Center, Dallas, United States; [3]Center for the Genetics of Host Defense, University of Texas Southwestern Medical Center, Dallas, United States; [4]Department of Molecular and Human Genetics, Baylor College of Medicine, Houston, United States; [5]Texas Children's Hospital, Houston, United States; [6]Institute of Human Genetics, University of Leipzig Medical Center, Leipzig, Germany; [7]The Saxon Epilepsy Center Kleinwachau, Radeberg, Germany; [8]Department of Genetics, King Faisal Specialist Hospital and Research Centre, Riyadh, Saudi Arabia; [9]Cellular and Molecular Research Center, Sabzevar University of Medical Sciences, Sabzevar, Islamic Republic of Iran; [10]Non-Communicable Diseases Research Center, Sabzevar University of Medical Sciences, Sabzevar, Islamic Republic of Iran; [11]Pardis Clinical and Genetics Laboratory, Mashhad, Islamic Republic of Iran; [12]Division of Human Genetics, Avicenna Research Institute, Mashhad University of Medical Sciences, Mashhad, Islamic Republic of Iran; [13]Greenwood Genetic Center, Greenwood, United States; [14]Department of Medical Genetics, University of British Columbia, British Columbia Children's and Women's Hospital Research Institute, Vancouver, Canada; [15]Department of Genetics, King Saud Medical City, Riyadh, Saudi Arabia; [16]Department of Pediatrics and Genetics, University of Colorado School of Medicine, Aurora, United States; [17]Department of Neuromuscular Diseases, University College London, Queen Square Institute of Neurology, London, United Kingdom; [18]Department of Psychiatry, University of Texas Southwestern Medical Center, Dallas, United States; [19]Peter O'Donnell Jr. Brain Institute, University of Texas Southwestern Medical Center, Dallas, United States

*For correspondence:
Maria.Chahrour@
UTSouthwestern.edu

Competing interests: The authors declare that no competing interests exist.

**Abstract** Autism spectrum disorder (ASD) is a constellation of neurodevelopmental disorders with high phenotypic and genetic heterogeneity, complicating the discovery of causative genes. Through a forward genetics approach selecting for defective vocalization in mice, we identified *Kdm5a* as a candidate ASD gene. To validate our discovery, we generated a *Kdm5a* knockout mouse model (*Kdm5a*-/-) and confirmed that inactivating *Kdm5a* disrupts vocalization. In addition, *Kdm5a*-/- mice displayed repetitive behaviors, sociability deficits, cognitive dysfunction, and abnormal dendritic morphogenesis. Loss of KDM5A also resulted in dysregulation of the

hippocampal transcriptome. To determine if *KDM5A* mutations cause ASD in humans, we screened whole exome sequencing and microarray data from a clinical cohort. We identified pathogenic *KDM5A* variants in nine patients with ASD and lack of speech. Our findings illustrate the power and efficacy of forward genetics in identifying ASD genes and highlight the importance of KDM5A in normal brain development and function.

## Introduction

Autism spectrum disorder (ASD) is characterized by deficits in communication, diminished social skills, and stereotyped behaviors. It is one of the most heritable of neuropsychiatric disorders (*Colvert et al., 2015*). Thousands of genetic variations, both common and rare, contribute to ASD risk (*Geschwind and State, 2015*). Causative genes identified to date encode molecules involved in a myriad of molecular pathways, with chromatin remodeling being one of the top pathways disrupted in ASD (*De Rubeis et al., 2014*). Due to the high phenotypic and genetic heterogeneity of ASD (*Betancur, 2011*; *Mitchell, 2011*), identification of genes underlying disease has been tedious and has required large-scale studies sequencing tens of thousands of patient samples (*Boyle et al., 2017*; *Li et al., 2013*). Despite recent strides being made in our understanding of the complex genetics of ASD, identified genes account for only ~30% of ASD cases with no single gene contributing to greater than 2% of cases (*de la Torre-Ubieta et al., 2016*; *Dias and Walsh, 2020*).

We utilized a forward genetics approach to rapidly identify new mutations causing ASD-related behaviors in mice. We screened mice bearing *N*-ethyl-*N*-nitrosourea (ENU)-induced mutations (*Wang et al., 2015*) for defects in ultrasonic vocalizations (USVs) and nest-building behavior. Automated meiotic mapping implicated *Kdm5a* as a candidate ASD gene and a nonsense mutation at that locus segregated with defective USVs and nest building. In addition to these phenotypes, targeted knockout of *Kdm5a* resulted in repetitive behaviors, deficits in social interaction, learning, and memory, and abnormal neuronal morphology. *KDM5A* encodes a chromatin regulator that belongs to the KDM5 family of lysine-specific histone H3 demethylases. Histone H3 lysine 4 trimethyl (H3K4me3) marks are present at gene promoters and active enhancers, and are regulated by multiple factors, including the KDM5 family members KDM5B and KDM5C (*Shen et al., 2017*). Mutations in lysine demethylases that regulate the demethylation of the H3K4me3 marks have been strongly linked to a host of neurodevelopmental disorders. Finally, we identified nine patients with ASD and lack of speech who have pathogenic *KDM5A* variants.

## Results

### Forward genomics identifies *Kdm5a* as a candidate ASD gene

To identify candidate ASD genes, we screened ENU mutagenized mice for ASD-like behavioral phenotypes. Specifically, we screened for abnormalities in USVs (*Portfors, 2007*) and in nest-building behavior (*Crawley, 2004*). USV quantification is a well-documented tool to assess social communication in mice (*Scattoni et al., 2009*; *Branchi et al., 2001*), and USV deficits are indicators of abnormal brain development. Mouse models of neurodevelopmental disorders, including ASD models, tend to vocalize less frequently or display abnormal vocalization parameters compared to wild type (WT) mice (*Lai et al., 2014*). In addition, the ability to build a communal nest is an indicator of sociability in mice (*Crawley, 2007*). We performed ENU mutagenesis in male C57BL/6J WT mice as previously described (*Wang et al., 2015*) and bred them to obtain ~30–50 third generation (G3) mice per pedigree (*Figure 1—figure supplement 1*). All G1 males were subjected to whole exome sequencing, and all G2 and G3 mice from each pedigree were genotyped for variants identified in the G1 pedigree founder.

We analyzed the ability of G3 mice to emit USVs at postnatal day (P) 4, scoring several different USV parameters (Materials and methods), and measured the quality of nests built at P29-32. For each USV parameter scored and for the nest-building behavior, mapping and linkage analysis was performed for every mutation, under the recessive, semidominant (additive), and dominant modes of inheritance, using Linkage Analyzer which detects phenovariance statistically linked to genotype as determined by a linear regression model (*Wang et al., 2015*). We screened a total of 350 G3 mice in eight pedigrees, that carried 409 mutations, and identified a causative allelic variant (*Selbst*)

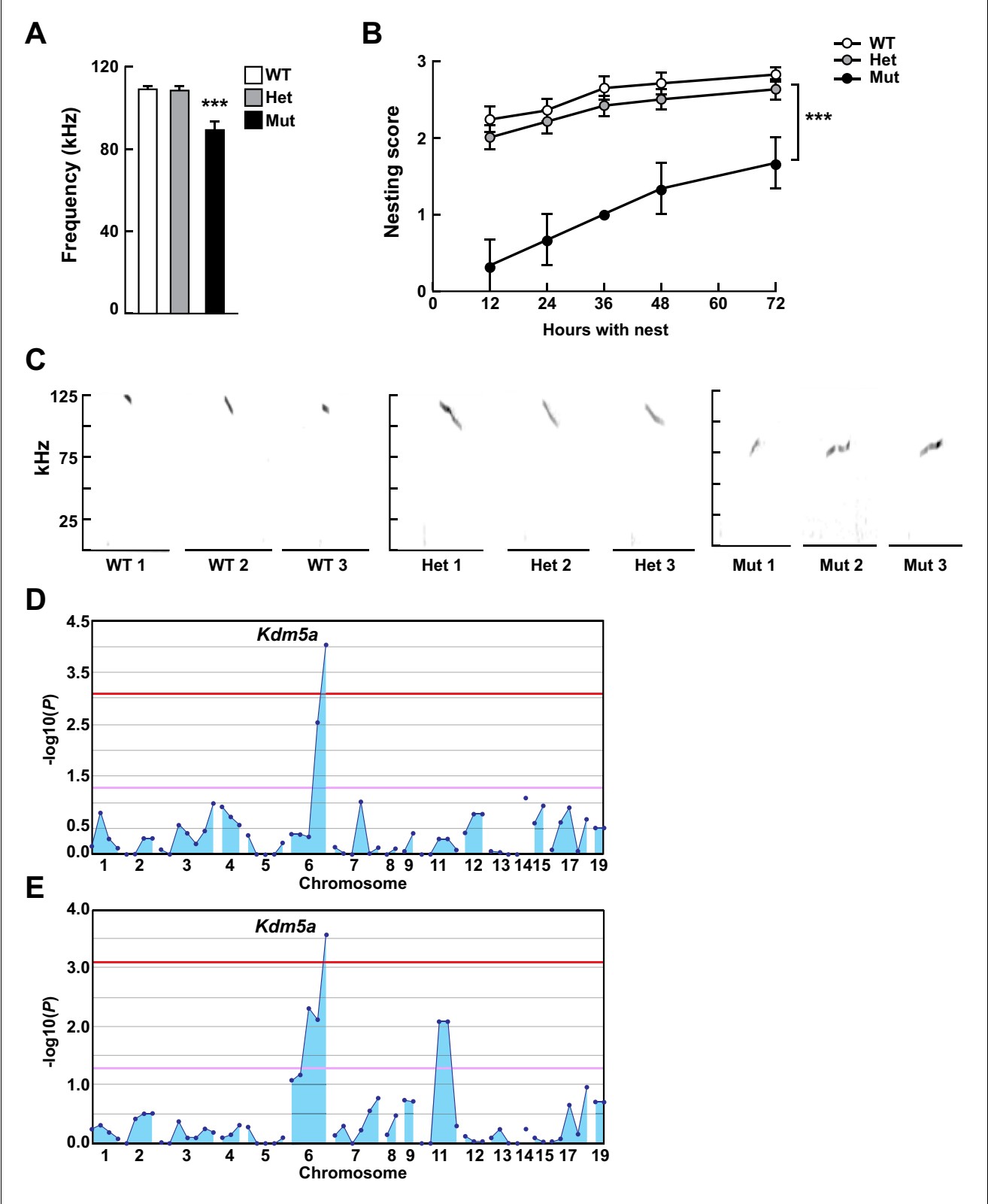

**Figure 1.** Mapping *Selbst*, an ASD-like phenotype of altered vocalization. (**A, B**) Quantitative phenotype data (maximum peak frequency (**A**) and nesting score (**B**)) were used for linkage analyses (n = 17 WT, 24 Het, 3 Mut). Corresponding phenotypic data plotted against genotype at the *Kdm5a* mutation site are shown. Values are mean ± SEM ((**A**) ***p=0.0001; (**B**) time p<0.0001, genotype ***p=0.0003, time x genotype p=0.5155). Data were analyzed using ordinary one-way ANOVA (**A**) or two-way ANOVA (**B**), followed by Tukey's multiple comparisons test. (**C**) Representative spectrograms

*Figure 1 continued*

of vocalizations from WT, heterozygous (Het), and homozygous (Mut) mutant mice (n = 3 WT, 3 Het, 3 Mut). (D, E) Linkage plots showing *P* values calculated using the recessive model for the ultrasonic vocalization (USV) (D) and the nesting (E) phenotypes. The -$\log_{10}$ *P* values (y axis) are plotted against the chromosomal positions of 60 mutations (x axis) identified in the G1 founder of the pedigree. Horizontal red and pink lines indicate thresholds of p=0.05 with or without Bonferroni correction, respectively.

The online version of this article includes the following figure supplement(s) for figure 1:

**Figure supplement 1.** Summary of the gene identification pipeline.
**Figure supplement 2.** *KDM5A* is ubiquitously expressed across tissues.

with abnormal USVs and nesting behavior (*Figure 1*). The *Selbst* phenotype showed clear recessive inheritance of reduced USV peak frequency (*Figure 1A and C*) and impaired nest-building ability (*Figure 1B* and *Figure 2—figure supplement 2B*). Whole exome sequencing and mapping identified a single peak on chromosome 6 linked to both the USV (p=2.9×10$^{-5}$) and the nesting (p=7.0×10$^{-5}$) phenotypes (*Figure 1D and E*). The peak corresponded to a single nucleotide substitution in *Kdm5a* resulting in a nonsense mutation (p.C322*) predicted to cause loss of the protein. *Kdm5a* encodes a chromatin regulator belonging to a family of histone lysine demethylases. It is expressed ubiquitously in humans and mice (*Smith et al., 2019*), including in the brain (*Figure 1— figure supplement 2*).

## Targeted *Kdm5a* knockout results in ASD-like behavioral phenotypes in mice

In order to investigate the physiological function of KDM5A and to genetically confirm that the *Kdm5a* nonsense mutation is the cause of the *Selbst* phenotype, we generated a *Kdm5a* constitutive knockout mouse model (*Kdm5a$^{-/-}$*). We targeted the mouse *Kdm5a* locus using CRISPR/Cas9 technology to insert a single base pair in exon 13, which was predicted to result in a frameshift mutation in codon 581 and to terminate translation after the inclusion of 8 aberrant amino acids (*Figure 2A*). We performed western blot analysis on cortical tissue and demonstrated loss of KDM5A in *Kdm5a$^{-/-}$* mice compared to WT and heterozygous littermates (*Figure 2B*).

*Kdm5a$^{-/-}$* mice were born in expected Mendelian ratios, exhibited normal body weight, body length, and brain to body weight ratio compared with control littermates (WT and *Kdm5a$^{+/-}$*) (*Figure 2—figure supplement 1*). To validate that the *Selbst* phenotype was due to loss of KDM5A, we analyzed the ability of the *Kdm5a$^{-/-}$* mice to emit USVs. We found that *Kdm5a$^{-/-}$* mice had reduced peak frequency of emitted USVs, which replicates the *Selbst* phenotype, and a severe reduction in the number of USVs (~84% decrease) compared with control littermates (*Figure 2C* and *Figure 2— figure supplement 2A*).

We assessed whether *Kdm5a$^{-/-}$* mice have ASD-like behavioral phenotypes. First, we characterized repetitive behaviors by measuring forepaw wringing and clasping through a modified tail suspension test (*Gandal et al., 2012*). We found that the *Kdm5a$^{-/-}$* mice spent more time wringing and clasping their forepaws compared with control littermates (*Figure 2D*, *Figure 2—video 1*, and *Figure 2—video 2*). We also measured self-grooming behavior and found that *Kdm5a$^{-/-}$* mice spent twice as much time self-grooming compared to control littermates (*Figure 2E*). Next, to asses gross locomotor activity, exploratory behavior, and anxiety, we performed the open field test. We noticed a slight increase in the total distance traveled by the *Kdm5a$^{-/-}$* mice, indicating a mild hyperactivity phenotype (*Figure 2F*). In addition, we found that the *Kdm5a$^{-/-}$* mice spent significantly less time in the center of the arena indicating a possible anxiety phenotype (*Figure 2F*). To assess any social behavior deficits, we measured direct social interaction (*Chang et al., 2017*; *Crawley, 2012*) and found that the *Kdm5a$^{-/-}$* mice spent significantly less time (~50%) interacting with novel partner mice compared to control littermates (*Figure 2G*). We then conducted the Morris water maze test to analyze the ability of *Kdm5a$^{-/-}$* mice to learn and memorize. WT and *Kdm5a$^{+/-}$* littermates had normal learning abilities shown by the decrease in latency to locate the hidden platform over a period of 4 days, as well as normal memory by showing preference for the quadrant where the platform was during the probe test (target quadrant). However, *Kdm5a$^{-/-}$* mice displayed severe disabilities in both learning and memory. Their latency to reach the hidden platform did not decrease over time and they did not show preference for the target quadrant during the probe test (*Figure 2H and I*).

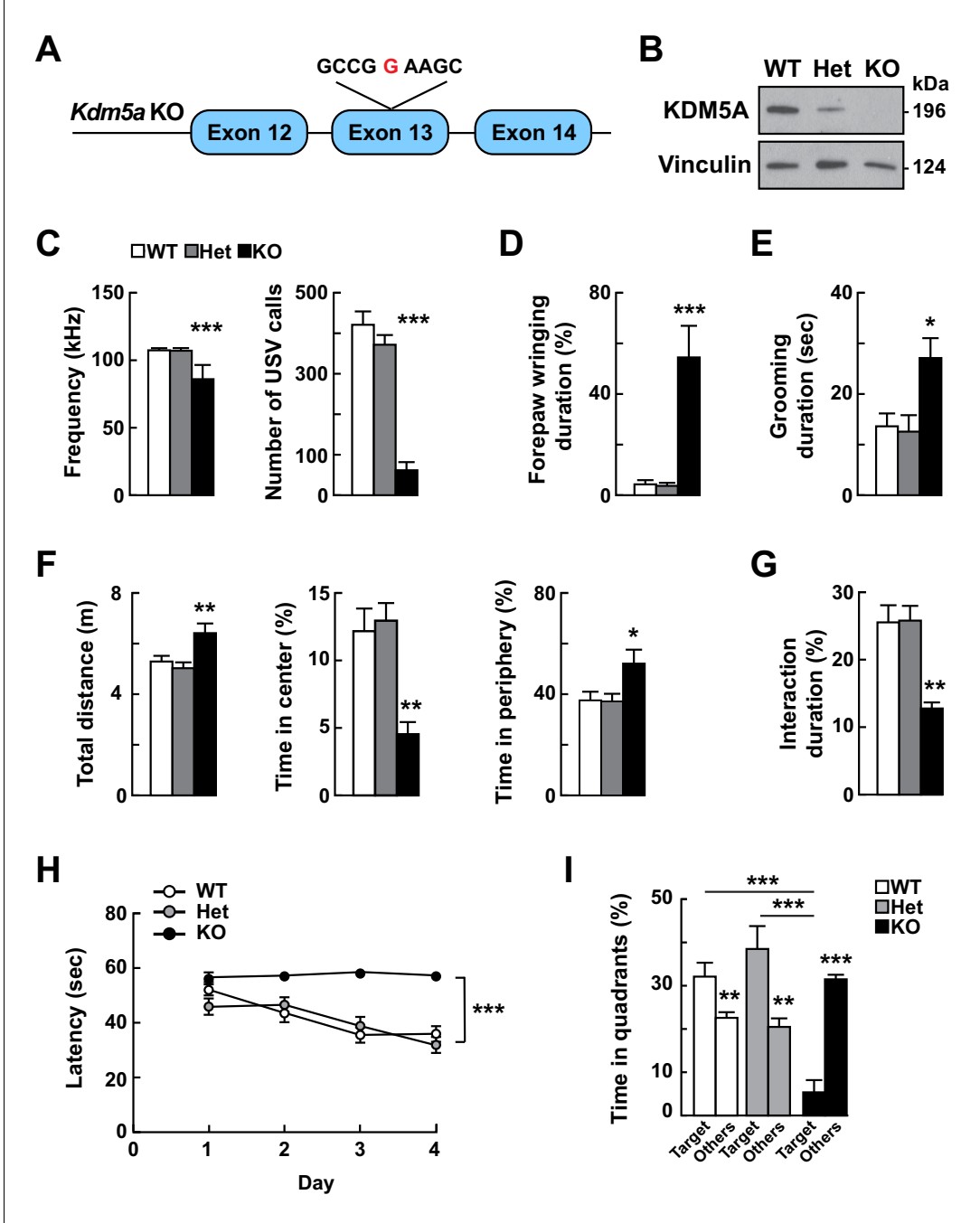

**Figure 2.** Loss of *Kdm5a* results in abnormal vocalizations, repetitive behaviors, and deficits in social behavior, learning, and memory. (**A**) Generation of *Kdm5a*$^{-/-}$ constitutive knockout (KO) mice. Schematic of the targeted *Kdm5a* locus disrupting exon 13 resulting in a frameshift and loss of KDM5A. (**B**) Western blot analysis from cortical tissue lysates of WT, *Kdm5a*$^{+/-}$ (Het), or *Kdm5a*$^{-/-}$ (KO) mice showed a ~ 50% decrease in KDM5A protein level in the Het and complete loss of KDM5A in the KO mice compared to WT. Representative immunoblots from three independent experiments (n = 3 WT, 3 Het, 3 KO). (**C**) KO mice have severe reduction in the maximum peak frequency of ultrasonic vocalizations (USVs) and in the number of USVs measured at P4 (***p<0.0001; n = 22 WT, 49 Het, 9 KO). (**D**) KO mice spent 60% of the time wringing and clasping their forepaws compared to 5% for WT and 4% for Het littermates (***p<0.0001; n = 25 WT, 25 Het, 10 KO). (**E**) KO mice spent more time self-grooming compared to WT and Het littermates (*p=0.0107; n = 6 WT, 7 Het, 5 KO). During the duration of the analysis, WT and Het mice had ~1-2 grooming events, while KO mice had ~3-4 grooming events. (**F**) KO mice traveled a longer distance, and spent less time in the center and more time in the periphery of the open field compared to WT and Het littermates (total distance: **p=0.0014, time in center: **p=0.0015, time in periphery: *p=0.0143; n = 13 WT, 13 Het, 7 KO). (**G**) KO mice showed decreased social interaction with a novel partner mouse compared to WT and Het littermates (**p=0.0021; n = 11 WT, 8 Het, 6 KO). For (**C**), (**D**), (**E**), (**F**), and (**G**) data were analyzed using ordinary one-way ANOVA followed by Tukey's multiple comparisons test. (**H**) Compared to their control

*Figure 2 continued on next page*

*Figure 2 continued*

littermates, KO mice have an impaired learning shown by the latency to locate the hidden platform that does not decrease during the training phase of the Morris water maze task (day p<0.0001, genotype ***p<0.0001, day x genotype p=0.0012; n = 13 WT, 10 Het, 6 KO). Data were analyzed using two-way ANOVA followed by Tukey's multiple comparisons test. (I) KO mice performed poorly in the Morris water maze probe test compared to their control littermates and spent more time in non-target quadrants (Others) (WT: **p=0.0057, Het: **p=0.0034, KO: ***p<0.0001, target WT vs target KO: ***p=0.0004, target Het vs target KO: ***p<0.0001, target WT vs target Het: p=0.4433; n = 13 WT, 10 Het, 6 KO). Data were analyzed using unpaired t test for the within-genotype analyses and ordinary one-way ANOVA followed by Tukey's multiple comparisons test for the across-genotype analyses. All behaviors, except USV analysis, were assessed at 5–9 weeks of age. All values are mean ± SEM.

The online version of this article includes the following video and figure supplement(s) for figure 2:

**Figure supplement 1.** Loss of *Kdm5a* does not affect growth.

**Figure supplement 2.** Altered vocalizations following loss of *Kdm5a*.

**Figure 2—video 1.** Related to *Figure 2*.

https://elifesciences.org/articles/56883#fig2video1

**Figure 2—video 2.** Related to *Figure 2*.

https://elifesciences.org/articles/56883#fig2video2

## Abnormal dendritic morphogenesis of cortical neurons in *Kdm5a* knockout mice

To assess the role of KDM5A in neuronal development, we measured dendritic complexity, length, and spine density *in vivo* by Golgi-Cox staining of brains from *Kdm5a*$^{-/-}$ and WT littermates. Sholl analysis revealed a severe reduction (~67%) in dendritic complexity of cortical neurons from *Kdm5a*$^{-/-}$ mice (*Figure 3A and B*). Furthermore, cortical neurons from *Kdm5a*$^{-/-}$ mice had significant reduction in dendritic length (~50% decrease) and dendritic spine density (~33% decrease) compared with neurons from WT littermates (*Figure 3C and D*).

### *KDM5A* mutations in patients with ASD

The human homolog, *KDM5A*, is extremely intolerant to loss-of-function variation based on data from the Genome Aggregation Database (gnomAD) (probability of loss-of-function intolerance (pLI) = 1.00) (*Genome Aggregation Database Consortium et al., 2020*). Prior to our study, it was not known whether *KDM5A* loss-of-function mutations result in ASD or in other neurodevelopmental disorders. To date only a single recessive missense variant in *KDM5A* has been reported to be associated with an undefined developmental disorder (intellectual disability and facial dysmorphisms) in one family (*Najmabadi et al., 2011*). It is worth noting that genes encoding other members of the KDM5 family, *KDM5B and KDM5C*, are disrupted in neurodevelopmental disorders (*De Rubeis et al., 2014*; *Lebrun et al., 2018*; *Al-Mubarak et al., 2017*; *Adegbola et al., 2008*; *Vallianatos et al., 2018*).

Through the forward genetics approach and the targeted mouse model, we identified a role for *Kdm5a* in regulating mouse vocalizations, nesting, repetitive behaviors, social interaction, cognitive function, and neuronal morphogenesis. Based on our findings, we sought to identify individuals with pathogenic *KDM5A* mutations and neurobehavioral phenotypes. Through global collaborative efforts, we identified nine individuals from seven families where clinical whole exome sequencing or chromosomal microarray analysis revealed pathogenic variants in *KDM5A* that segregated with phenotype in these families (*Figure 4A* and Clinical presentation). Apart from the *KDM5A* variants, no variants were identified in known disease-causing genes that accounted for the clinical phenotype of the nine individuals. Three individuals carried homozygous deletions removing four exons (in two individuals) or ten exons (in one individual) of *KDM5A*, three individuals carried homozygous recessive variants (two splice site and one missense), and the remaining three individuals were heterozygous for *de novo* variants (two missense and one nonsense) (*Figure 4B*, *Figure 4—figure supplement 1*, *Figure 4—figure supplement 2*, and *Table 1*). All the identified variants were absent from exome or genome sequencing data of 141,456 unrelated individuals in gnomAD (*Genome Aggregation Database Consortium et al., 2020*), confirming their rarity.

All nine individuals with *KDM5A* mutations presented clinically with ASD and a spectrum of neurodevelopmental phenotypes including complete lack of speech, intellectual disability, and developmental delay (*Table 1* and Clinical presentation). The proband from the first family (KD-1–3) was heterozygous for a missense mutation in the first methionine of KDM5A (p.Met1Leu), which we

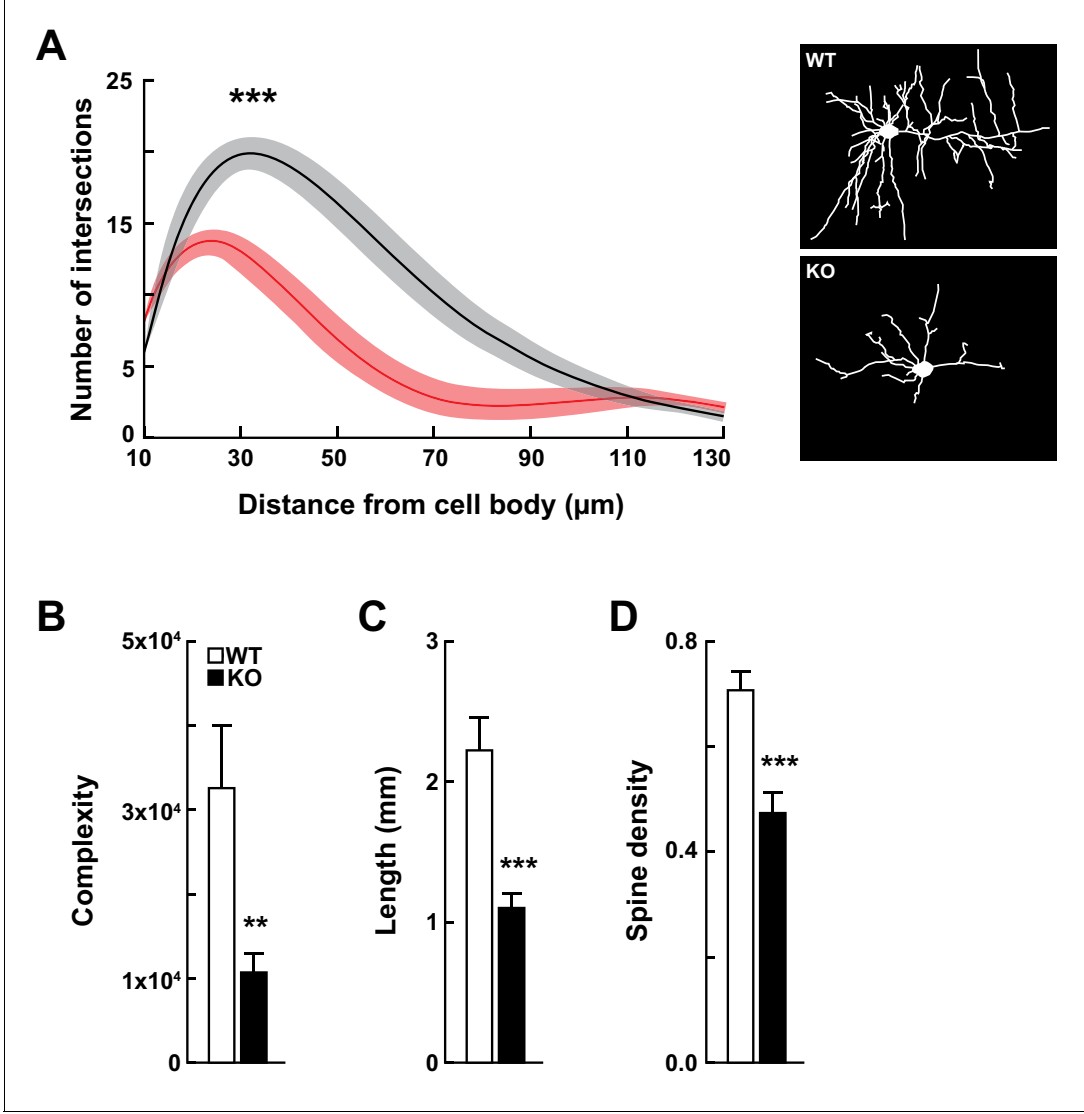

**Figure 3.** *Kdm5a* knockout mice have impaired dendritic morphogenesis. (**A**) Sholl analysis from Golgi-Cox-stained neurons revealed a reduction in dendritic complexity of cortical neurons from KO mice (red) compared to WT (black) littermates (***p<0.0001). Data were analyzed using two-way ANOVA followed by Tukey's multiple comparisons test. (Right) Representative tracings of Golgi-Cox-stained cortical layer II/III neurons. Golgi-Cox staining showed significantly reduced dendritic complexity (B, **p=0.0066), length (C, ***p<0.0001), and spine density (D, ***p<0.0001) of cortical neurons from KO mice compared to WT littermates. Data were obtained from basal dendrites of cortical layer II/III neurons from mice at 14–16 weeks of age. Data were analyzed using unpaired t test. All values are mean ± SEM (n = 15 WT, 15 KO).

validated via Sanger sequencing (*Figure 4—figure supplement 1B*). The mutation alters the highly conserved initiation methionine in KDM5A (*Figure 4—figure supplement 2A*). Western blot analysis of lymphoblastoid cell lines showed a reduction in KDM5A in the proband compared to the unaffected father (*Figure 4C*). Family KD-2 had two affected children homozygous for a ~ 10 kb deletion that removed exons 6 through 9 of *KDM5A* without disrupting the reading frame (*Figure 4—figure supplement 1A*). Western blot analysis of lymphoblastoid cell lines revealed a truncated protein of the predicted molecular weight (174 kDa) in the proband compared to the unaffected parents (*Figure 4C*). The proband in the third family (KD-3–3) was heterozygous for a missense mutation in a highly conserved arginine residue of KDM5A (p.Arg1428Leu) (*Figure 4—figure supplement 2B*). The two affected children in family KD-4, who were offspring of a consanguineous mating, presented with the most severe phenotype that appeared to worsen with age given the phenotype of the older

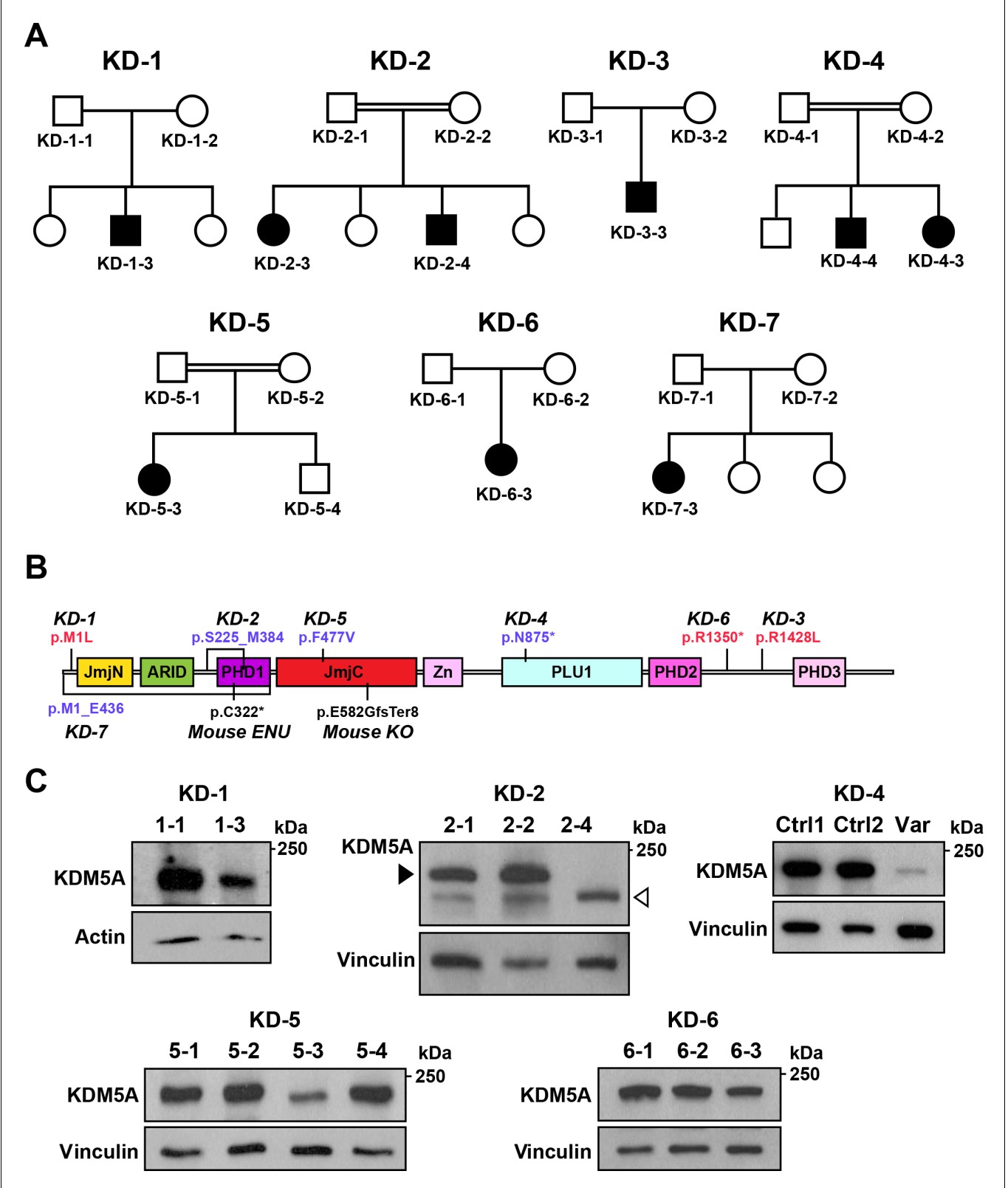

**Figure 4.** Identification of *KDM5A* mutations in patients with ASD. (**A**) Pedigrees of seven families with *KDM5A* mutations in nine probands with ASD. Double lines: first cousin status; circles: females; squares: males; shaded symbols: affected individuals. (**B**) A schematic of KDM5A domains and location of the identified mutations. *De novo* and recessive mutations are indicated in red and blue, respectively. The *Selbst* mutation is a cysteine to premature stop codon substitution at position 322 of the protein. ARID, A-T rich interaction domain; JmjC, Jumonji C; JmjN, Jumonji N; PHD, plant

*Figure 4 continued on next page*

Figure 4 continued

homeodomain; PLU1, putative DNA/chromatin binding motif; Zn, zinc finger. (C) Western blot analysis of lymphoblastoid cell line lysates from affected individuals (KD-1–3, KD-2–4, KD-5–3, and KD-6–3) and unaffected family members (KD-1–1, KD-2–1, KD-2–2, KD-5–1, KD-5–2, KD-5–4, KD-6–1, and KD-6–2) showed reduced KDM5A protein in affected individuals KD-1–3, KD-5–3, and KD-6–3, and a truncated KDM5A protein in affected individual KD-2–4. Western blot analysis of HEK293T cells with knock-in of the splice site mutation present in affected individuals KD-4–3 and KD-4–4 (Var) and HEK293T cells which underwent transfection but kept the reference sequence (Ctrl 1 and Ctrl 2), showed a decrease in KDM5A protein level in the targeted cells compared to control cells. β-actin and vinculin were used as loading controls. The black arrowhead points to the KDM5A band (196 kDa) and the white arrowhead points to the truncated KDM5A band (174 kDa).

The online version of this article includes the following video and figure supplement(s) for figure 4:

**Figure supplement 1.** Related to *Figure 4*.
**Figure supplement 2.** Related to *Figure 4*.
**Figure 4—video 1.** Related to *Figure 4*.
https://elifesciences.org/articles/56883#fig4video1
**Figure 4—video 2.** Related to *Figure 4*.
https://elifesciences.org/articles/56883#fig4video2

sibling (*Table 1*, Clinical presentation, *Figure 4—video 1*, and *Figure 4—video 2*). Both siblings carried a homozygous recessive variant disrupting the donor splice site of exon 18 (c.2541+1G > T). We introduced this mutation into the *KDM5A* locus in HEK293T cells using CRISPR/Cas9 (*Figure 4—figure supplement 1C*) and analyzed its impact on KDM5A using western blot analysis. We found that the splice site mutation resulted in a severe reduction in KDM5A in the targeted cells compared to control cells (*Figure 4C*). Another proband, KD-5–3, was homozygous for a missense mutation in a highly conserved residue in KDM5A (p.Phe477Val, *Figure 4—figure supplement 2C*) and resulted in a reduction in KDM5A in lymphoblastoid cells from the proband compared to the unaffected family members (*Figure 4C*). The proband in family KD-6 was heterozygous for a nonsense mutation in KDM5A that also caused a reduction in KDM5A in the proband compared to the unaffected parents (*Figure 4C*). In family KD-7, the proband was homozygous for a ~ 50 kb deletion that removed exons 1 through 10 of *KDM5A*. The deletion is predicted to result in loss of KDM5A transcript and protein.

## Impaired transcriptional profile in the hippocampus following loss of KDM5A

KDM5A is a histone lysine demethylase that removes methyl groups from lysine 4 of histone H3, resulting in repression of target genes. In order to identify KDM5A targets, we profiled transcripts from hippocampi of WT and *Kdm5a$^{-/-}$* mice. The RNA-seq data was of high quality and clustered robustly by genotype (*Figure 5—figure supplement 1* and *Figure 5—figure supplement 2A*). Loss of KDM5A resulted in dysregulation of 450 genes (FDR-corrected p≤0.05 and log$_2$ fold change ≥| 0.3|) (*Figure 5—source data 1*), of which 191 genes were upregulated and 259 genes were downregulated (*Figure 5A and B*). Interestingly, upregulated genes included those involved in the regulation of neurological processes (*Ednrb, Gba, Mtmr2, Nov, Npy2r*) and RNA splicing (*Cdc40, Dhx38, Lsm3, Lsm8, Sart3, Sfswap, Snrpb2, Tfip11, Uhmk1, Wbp11*), while downregulated genes included those involved in neurogenesis (e.g. *Myoc, Nphp1, Otx2, Sema3b, Sgk1, Six3os1, Trpv4*) and cell proliferation (e.g. *Ace, Adora2a, Aqp1, Cdh3, Enpp2, Nde1*) (*Figure 5—figure supplement 2B* and *Figure 5—figure supplement 2—source data 1*). Expression of the other members of the KDM5 family, *Kdm5b, Kdm5c,* and *Kdm5d*, remained unchanged upon loss of *Kdm5a* (*Figure 5—figure supplement 3*). We confirmed the differential expression of the top upregulated gene, *Efcab6*, and the top downregulated gene, *Ptgds*, with qRT-PCR. We also confirmed the upregulation of *Shh*, known for its role in hippocampal neurogenesis, proliferation, and differentiation, and potential role in synaptic plasticity (*Yao et al., 2015*; *Yao et al., 2016*), and its downstream target *Ccnd1*, a mediator of cell cycle progression (*Figure 5C*). The expression of homologs of several known ASD genes was altered upon loss of *Kdm5a*: *Ccnk, Cnr1, Hsd11b1, Oxtr, Ptprb,* and *Styk1* were upregulated, whereas *Ace, Adora2a, Dmpk, Mfrp, Myo5c, Ppp1r1b, Setdb2, Slc29a4,* and *Stk39* were downregulated (*Figure 5D*). To further emphasize the importance of KDM5A in brain development, we compared the set of genes dysregulated in the *Kdm5a$^{-/-}$* hippocampus to genes involved in neurodevelopment based on data from the BrainSpan Atlas of the Developing Human Brain

**Table 1.** Clinical phenotype of patients with *KDM5A* mutations identified in this study.

All findings reported at latest exam. Mutations are reported on human genome GRCh37/hg19 coordinates. MAF, minor allele frequency in gnomAD; NP, not present; UK, unknown. * Proband received extensive speech therapy since early childhood.

| Patient | KD-1-3 | KD-2-3 | KD-2-4 | KD-3-3 | KD-4-3 | KD-4-4 | KD-5-3 | KD-6-3 | KD-7-3 |
|---|---|---|---|---|---|---|---|---|---|
| Sex | Male | Female | Male | Male | Female | Male | Female | Female | Female |
| Age at latest evaluation (years) | 5 | 12 | 8 | 18 | 3 | 20 | 4 | 40 | 13 |
| Nucleotide change (NM_001042603) | c.1A>T | Deletion of exons 6 through 9 | Deletion of exons 6 through 9 | c.4283G>T | c.2541 +1G>T | c.2541 +1G>T | c.1429T>G | c.4048C>T | Deletion of exons 1 through 10 |
| Amino acid change | p.Met1Leu | — | — | p.Arg1428Leu | — | — | p.Phe477Val | p.Arg1350* | — |
| MAF (%) | NP | — | — | NP | NP | NP | NP | NP | — |
| Inheritance | *de novo* | Recessive | Recessive | *de novo* | Recessive | Recessive | Recessive | *de novo* | Recessive |
| Weight (percentile) | 95th | 90th | 99th | 1st | 84th | <1st | 50th | 10th | <1st |
| Height (percentile) | 98th | 96th | 99th | <1st | 99th | 7th | 30th | 25th | <1st |
| Head circumference (percentile) | 50th | <5th | 99th | UK | 50th | <5th | <5th | <3rd | <2nd |
| ASD | + | + | + | + | + | + | + | UK | UK |
| Absent speech | + | + | + | - * | + | + | + | Few words | + |
| Intellectual disability | + | + | + | + | + | + | + | + | + |
| Developmental delay | + | + | + | + | - | + | + | + | + |
| Seizures | + | + | - | + | - | - | - | + | - |
| Motor impairment | + | + | + | - | + | + | + | + | + |
| Muscle hypotonia | + | + | + | - | - | + | + | - | + |
| Feeding difficulties | + | + | + | + | - | + | + | - | - |
| Facial dysmorphisms | - | + | + | + | - | - | + | + | + |
| Abnormal MRI | UK | Hypoplastic corpus callosum | Hypoplastic corpus callosum, mild hippocampal atrophy | UK | - | UK | Periventricular leukomalacia | Gliosis of the parieto-central pyramidal tracts and mild atrophy of the parietal region | UK |
| Cardiac defects | Murmur | Atrial septal defect | - | Ventricular septal defect | - | - | Atrial septal defect | - | Murmur, coarctation of the aorta |

(*Miller et al., 2014*). We found that several of the genes dysregulated in *Kdm5a*$^{-/-}$ mice are homologs of essential neurodevelopmental genes (*Figure 5E*).

## Discussion

Forward genetics has long been recognized as an unbiased genome-wide method to identify genes underlying many biological processes (*Beutler, 2016*). By screening ENU mutagenized mice for ASD-like behaviors, we identified *Kdm5a* as a candidate ASD gene that regulates vocalization and nesting. To our knowledge, this is the first successful identification of an ASD gene through a

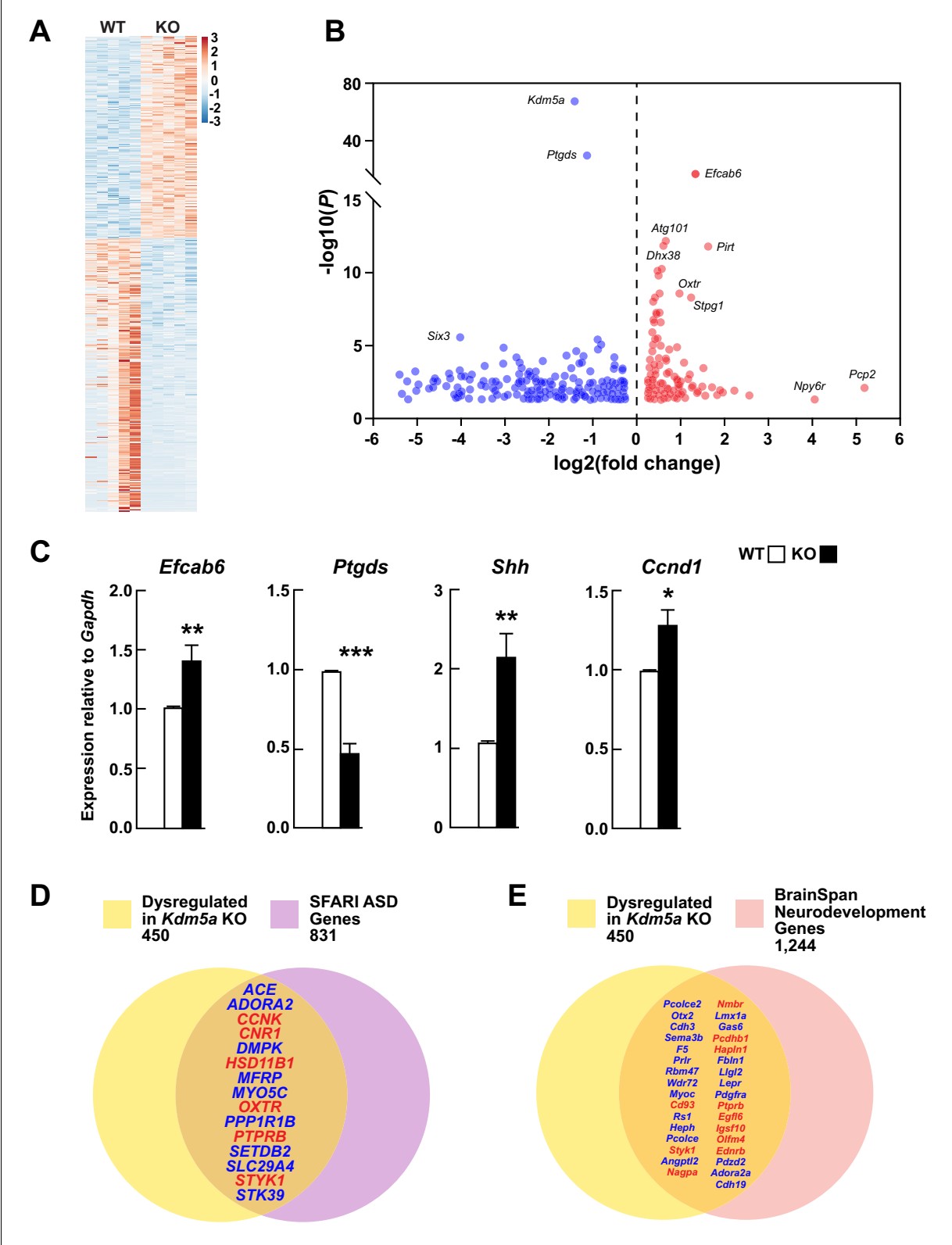

**Figure 5.** Transcriptional dysregulation in hippocampi of *Kdm5a* knockout mice. (**A**) Heatmap showing hippocampal gene expression profiles in WT and *Kdm5a^-/-* (KO) mice at 6–8 weeks of age (n = 5 WT, 5 KO). Red and blue indicate upregulation and downregulation, respectively. Data plotted from genes with a log$_2$ fold change ≥ |0.3| and FDR-corrected p≤0.05. (**B**) Volcano plot of the RNA-seq data showing the log$_2$ fold change in gene expression in *Kdm5a^-/-* compared to WT hippocampi. (**C**) qRT-PCR gene expression validation of the most upregulated and downregulated genes in

*Figure 5 continued on next page*

Figure 5 continued

*Kdm5a^{-/-}* hippocampi, *Efcab6* and *Ptgds*, respectively, as well as upregulation of *Shh* and its downstream target *Ccnd1*. Values are mean ± SEM (*Efcab6*: **p=0.0084, *Ptgds*: ***p<0.0001, *Shh*: **p=0.0056, *Ccnd1*: *p=0.0114; n = 5 WT, 5 KO). Data were analyzed using unpaired t test. Genes dysregulated following loss of *Kdm5a* overlap with homologs of known ASD genes (D) and of genes underlying neurodevelopment (E). Red and blue indicate upregulated and downregulated genes, respectively.

The online version of this article includes the following source data and figure supplement(s) for figure 5:

**Source data 1.** Genes dysregulated in the hippocampus of *Kdm5a* KO mice.
**Figure supplement 1.** Quality control metrics of RNA-seq reads as determined by FastQC and STAR.
**Figure supplement 2.** Related to *Figure 5*.
**Figure supplement 2—source data 1.** Gene ontology analysis of genes dysregulated in the hippocampus of *Kdm5a^{-/-}* mice.
**Figure supplement 3.** Expression levels of *Kdm5b*, *Kdm5c*, and *Kdm5d* remain unchanged following loss of *Kdm5a*.

forward genetics strategy. In our screen, we observed damaging homozygous mutations in ~0.2% of all genes in three or more mice, and we were able to successfully identify a candidate ASD gene, indicating the efficiency of this strategy for highly heterogeneous disorders. We generated a *Kdm5a* knockout mouse model in order to validate the *Selbst* phenotype and to characterize the function of KDM5A. We found that *Kdm5a^{-/-}* mice indeed showed significant alterations in their emitted USVs indicating abnormal brain development upon loss of *Kdm5a*. Furthermore, *Kdm5a^{-/-}* mice showed increased self-grooming and remarkable repetitive forepaw wringing and clasping, a characteristic ASD-like phenotype observed in mouse models of several neurodevelopmental disorders, one of the most well-characterized examples being mouse models of Rett syndrome (*Chao et al., 2010*). *Kdm5a^{-/-}* mice also had mild hyperactivity and showed signs of anxiety in the open field test. Additional behavioral characterization is required to thoroughly assess anxiety in the *Kdm5a^{-/-}* mice. Furthermore, *Kdm5a^{-/-}* mice displayed social behavior deficits and abnormalities in learning and memory. In addition, loss of KDM5A resulted in severe neuronal morphogenesis abnormalities, which included a decrease in dendritic complexity, length, and spine density. These findings reveal an essential role for KDM5A in dendritic morphogenesis and maturation, processes essential for the establishment of neuronal circuitry and wiring of the brain (*Dong et al., 2015*). Collectively, the neurobehavioral phenotypes observed in the *Kdm5a^{-/-}* mice uncover the function of KDM5A in the brain and recapitulate phenotypes observed in patients with *KDM5A* mutations, including deficits in sociability, cognitive function, and repetitive behaviors.

Chromatin regulation is essential for gene expression and brain development. Mutations in several genes coding for chromatin remodelers and epigenetic modulators have been linked to neurodevelopmental disorders including ASD, with causative mutations identified in several chromatin remodelers (e.g. *CHD8* and *ARID1B*) (*O'Roak et al., 2012*; *Turner et al., 2016*). Moreover, two genes in the KDM5 family, *KDM5B* and *KDM5C*, are disrupted in neurodevelopmental disorders, including intellectual disability and ASD (*De Rubeis et al., 2014*; *Iossifov et al., 2014*; *Jensen et al., 2005*). This is the first report implicating disruption of *KDM5A* as a cause of ASD. The fact that mutations in *KDM5A*, *KDM5B*, and *KDM5C* all cause neurodevelopmental disorders, indicates that their function in the brain is nonredundant.

We found seven pathogenic *KDM5A* variants in patients with ASD, in conjunction with lack of speech, intellectual disability, and developmental delay. Despite the phenotypic variability typically observed within the different genetic subtypes of ASD, all the patients with *KDM5A* mutations reported in our study have a complete absence of speech, suggesting a potential role for KDM5A in regulating social communication in humans. Both inherited and *de novo* mutations were identified in *KDM5A*, similar to many other ASD genes including *KDM5B* (*Lebrun et al., 2018*), *CHD8* (*Guo et al., 2018*), and *DOCK8* (*De Rubeis et al., 2014*; *Guo et al., 2018*). We analyzed the impact of the *KDM5A* mutations and found that they resulted in decreased protein levels and in one case a truncated protein. The recessive missense mutation in KD-5 results in a single amino acid substitution in one of the enzymatic domains of KDM5A, the Jumanji C (JmjC) domain (*Pilka et al., 2015*; *Horton et al., 2016*). The homozygous deletion in KD-2 results in loss of the plant homeodomain 1 (PHD1), which binds unmodified H3K4 (*Torres et al., 2015*) and was shown to enhance the demethylase catalytic activity (*Longbotham et al., 2019*). Furthermore, the fact that the KD-2 deletion results in a truncated protein which is also present in the healthy carrier parents argues against the mutation being dominant negative. Further functional studies are needed to understand how the identified

mutations may affect KDM5A localization, enzymatic function, chromatin binding ability, or interaction(s) with binding partners. The fact that we identified both monoallelic as well as biallelic mutations in *KDM5A* emphasizes the sensitivity of the brain to KDM5A levels.

KDM5A demethylates H3K4me3, a histone mark known to be associated with memory formation (*Collins et al., 2019a*; *Gupta et al., 2010*) and memory retrieval (*Webb et al., 2017*). Perturbations in histone demethylases and methyltransferases at this mark give rise to neurodevelopmental disorders, including ASD and intellectual disability (*Vallianatos and Iwase, 2015*), emphasizing the importance of H3K4me3 in learning and memory (*Collins et al., 2019b*). Thus, we analyzed changes in the hippocampal transcriptome upon loss of KDM5A. By demethylating H3K4me3, an activating histone mark, KDM5A would be expected to repress the expression of its target genes, suggesting that genes upregulated in the hippocampus of the $Kdm5a^{-/-}$ mice are potential direct targets of KDM5A. Genes upregulated in $Kdm5a^{-/-}$ clustered in regulation of neurological processes and RNA splicing, suggesting a role for KDM5A in these pathways. In addition, *Shh*, which plays an essential role in hippocampal neurogenesis and neural progenitor cell proliferation and differentiation, and its downstream target *Ccnd1*, were also upregulated in $Kdm5a^{-/-}$, indicating a potential mechanism by which KDM5A could be exerting its function in the hippocampus. Downregulated genes were involved in neurogenesis and cell proliferation, suggesting that loss of KDM5A altered these processes that are essential for normal hippocampal function. The most upregulated gene, a potential direct target of KDM5A, was *Efcab6* (EF-Hand Calcium Binding Domain 6; $\log_2$ fold change = 1.34, FDR-corrected p=$7.68\times10^{-18}$), which represses transcription of the androgen receptor by recruiting histone deacetylase complexes (*Niki et al., 2003*). The most downregulated gene was *Ptgds* (Prostaglandin D2 Synthase; $\log_2$ fold change = $-1.13$, FDR-corrected p=$1.13\times10^{-30}$), which functions as a neuromodulator and a trophic factor in the brain (*Taniguchi et al., 2007*). Out of 450 differentially expressed genes, 15 were homologs of known ASD genes, including *Ace*, *Cnr1*, *Mfrp*, *Oxtr*, and *Ppp1r1b*, and 33 were homologs of genes essential for brain development, including *Cdh3*, *Otx2*, *Pcolce2*, and *Sema3b*, highlighting the importance of KDM5A in brain development and function. Furthermore, we did not observe an increase in the expression of *Kdm5b*, *Kdm5c*, or *Kdm5d* in the $Kdm5a^{-/-}$ hippocampus, suggesting that there is no compensation from other KDM5 family members following loss of KDM5A. Others have made a similar observation where transcript levels of KDM5 family members were unchanged following loss of *Kdm5b* (*Albert et al., 2013*) and *Kdm5c* (*Iwase et al., 2016*) in neurons and brain tissue. Taken together, our transcriptomics data identify many neuronal genes perturbed in the hippocampus of $Kdm5a^{-/-}$ mice, and point to potential pathways through which KDM5A mediates its function in the brain.

In summary, using a forward genetics strategy, we identified a new gene, *KDM5A*, that is disrupted in patients with ASD and lack of speech. We further uncovered an essential role for KDM5A in brain development and function. Our approach underscores the value of leveraging forward genetics in mice to discover new gene defects that cause phenotypically complex and genetically heterogeneous disorders. Future studies dissecting the function of KDM5A in the developing brain will elucidate the molecular underpinnings of this subtype of ASD, and more broadly inform us on the role of chromatin remodeling in neurodevelopment and disease.

# Materials and methods

## Key resources table

| Reagent type (species) or resource | Designation | Source or reference | Identifiers | Additional information |
|---|---|---|---|---|
| Antibody | 'Mouse monoclonal' anti-β-Actin | Abcam | # ab6276; RRID:AB_2223210 | WB (1:1000) |
| Antibody | 'Rabbit polyclonal' anti-KDM5A | Abcam | # ab70892; RRID:AB_2280628 | WB (1:1000) |
| Antibody | 'Rabbit polyclonal' anti-Vinculin | Cell Signaling | # 4650S; RRID:AB_10559207 | WB (1:1000) |
| Blood samples (*Homo sapiens*) | | Referring clinicians | N/A | |

*Continued on next page*

*Continued*

| Reagent type (species) or resource | Designation | Source or reference | Identifiers | Additional information |
|---|---|---|---|---|
| Cell line (*H. sapiens*) | Lymphoblastoid cell lines | UTSW Human Genetics Clinical Laboratory | N/A | |
| Cell line (*H. sapiens*) | HEK293T | ATCC | # CRL-3216; RRID:CVCL_0063 | |
| Strain, strain background (*Mus musculus*) | C57BL/6N | Charles River Laboratories | Strain Code: 027 | Wild type mice |
| Strain, strain background (*M. musculus*) | C57BL/6J | In-house | In-house | *Kdm5a*$^{-/-}$ CRISPR/Cas9 mice |
| Sequence-based reagent | *Kdm5a*$^{-/-}$ CRISPR/Cas9 sgRNA | Integrated DNA Technologies | N/A | TTAATACGACTCACTATAG GGGATACAACTTTGCCGAAG |
| Sequence-based reagent | Shh_F | Integrated DNA Technologies | qPCR primer | ACTGGGTCTACTATGAATCC |
| Sequence-based reagent | Shh_R | Integrated DNA Technologies | qPCR primer | GTAAGTCCTTCACCAGCTTG |
| Sequence-based reagent | Ccnd1_F | Integrated DNA Technologies | qPCR primer | TTCCCTTGACTGCCGAGAAG |
| Sequence-based reagent | Ccnd1_R | Integrated DNA Technologies | qPCR primer | AAATCGTGGGGAGTCATGGC |
| Sequence-based reagent | Efcab6_F | Integrated DNA Technologies | qPCR primer | CTGGAGCAGTGAGGGTCAAC |
| Sequence-based reagent | Efcab6_R | Integrated DNA Technologies | qPCR primer | ATGGTCCCCGTGTCCCTAAG |
| Sequence-based reagent | Ptgds_F | Integrated DNA Technologies | qPCR primer | GCTCCTTCTGCCCAGTTTTC |
| Sequence-based reagent | Ptgds_R | Integrated DNA Technologies | qPCR primer | CCCCAGGAACTTGTCTTGTTGA |
| Recombinant DNA reagent | pU6-(BbsI)_CBh-Cas9-T2A-mCherry (plasmid) | Addgene | RRID:Addgene_64324 | Plasmid for CRISPR knock-in |
| Software, algorithm | Adobe Illustrator | Adobe | RRID:SCR_010279 | |
| Software, algorithm | GraphPad Prism | GraphPad | RRID:SCR_002798 | |
| Software, algorithm | Avisoft-RECORDER | Avisoft Bioacoustics | RRID:SCR_014436 | |

## Subjects and specimens

The study was approved by the University of Texas Southwestern Medical Center (UTSW) Institutional Review Board (protocol number STU 032015–014). Seven families (KD-1 through KD-7) with ASD and *KDM5A* mutations (*Table 1*) were identified through GeneDx and GeneMatcher (*Sobreira et al., 2015*), and enrolled in our study after obtaining written informed consent from all study participants. Subjects were evaluated in a clinical setting and phenotypes of the affected individuals are detailed below under Clinical presentation. Peripheral blood samples were collected from the affected individuals and family members. Genomic DNA was isolated from circulating leukocytes using AutoPure (Qiagen) according to the manufacturer's instructions. *KDM5A* variants were identified as detailed below for each family, either through whole exome or genome sequencing or microarray analysis. Lymphocytes were isolated and Epstein-Barr virus-transformed lymphoblastoid cell lines were generated at the UTSW Human Genetics Clinical Laboratory, for families KD-1, KD-2, KD-5, and KD-6.

## Clinical presentation
### Family KD-1

The first proband (KD-1–3) is a boy from a United States family of European and Hispanic ancestry. The proband has two unaffected female siblings. He presented with ASD, absent speech, moderate to severe intellectual disability, and significant developmental delays. On physical exam at age five his growth parameters were normal and he had normal facial features with carious teeth. He is

unable to feed himself and has feeding difficulties requiring a high calorie special formula. He is still in diapers and has sleep problems, in part related to low ferritin, for which he required an intravenous iron infusion. He attends a special school for children with ASD and is receiving applied behavioral analysis (ABA) therapy. He is very aggressive toward other family members and his classmates. Other findings include a flow murmur, tight phimosis, moderate pectus excavatum, a mild increase in joint range of motion, and slightly brisk deep tendon reflexes in his lower extremities. Clinical whole exome sequencing (WES) was performed at GeneDx. The identified variant was validated through targeted Sanger sequencing.

## Family KD-2

The probands (KD-2–3 and KD-2–4) are siblings from a consanguineous Canadian family of Afghani ancestry. The female proband (KD-2–3) presented to a pediatric hospital at 4 months of age with failure to thrive. The identified anomalies included: global developmental delay, hypotonia, feeding difficulty, gastroesophageal reflux, small accessory loop of the small bowel, anterior segment dysgenesis of the right ocular globe with an eccentric pupil, posterior embryotoxon, and iris hypoplasia, borderline microcephaly, atrial septal defect, hypoplastic and dysgenic corpus callosum, and unusual morphological features including hypertelorism and a prominent glabellar region. She was a first pregnancy for her healthy parents. Follow-up over time demonstrated limited developmental progress with ongoing dependence on gastrostomy feeding. She required surgery for strabismus and for repair of her heart defect. Nerve conduction studies were normal. Extensive metabolic screening, including analysis of the cerebrospinal fluid, was non-diagnostic. Genetic investigations included karyotyping, chromosomal microarray analysis, and later exome sequencing, none of which yielded a candidate cause for her condition. By 3 years of age she could ambulate with difficulty with her hands held, and she was diagnosed with cerebral palsy. She has remained non-verbal, and at 5 years of age was diagnosed with ASD on the basis of her limited social interaction and motor stereotypies, including hand-wringing and bruxism, as well as severe intellectual disability. She developed focal seizures at 9 years of age controlled with Lamotrigine.

The second child born to the parents was unaffected. The third child (KD-2–4), a male, was found at birth to have hypotonia, feeding difficulties, hypertelorism, and a prominent glabellar region. Growth parameters were normal. Brain magnetic resonance imaging (MRI) identified a dysgenic corpus callosum. He also required surgery for strabismus. He had profound global developmental delay and was diagnosed with ASD, intellectual disability, and cerebral palsy. He has remained non-ambulatory and non-verbal at 7 years of age. A fourth child was born who was unaffected.

The family enrolled in a research study, approved by the institutional review board of the University of British Columbia (H15-00092) for whole genome sequencing, which was performed with an Illumina platform at the British Columbia Genome Sciences Centre on both parents and affected siblings. A combination of read depth calling and split-read analysis identified a homozygous deletion encompassing four exons of *KDM5A*, spanning 10 kb on chr12:460,661–470,642. There were no other compelling candidate variants. The copy number variant was confirmed with chromosomal microarray for the four family members, which was also performed for the two unaffected siblings who were heterozygous carriers like their parents.

## Family KD-3

The proband (KD-3–3) is an 18 year old male from a United States family of European ancestry. He was first evaluated at 5 years of age and diagnosed with ASD, echolalia, and intellectual disability. He received physical, speech, and occupational therapies for many years. He was born at 34 weeks of gestation weighing 1.38 kg. The pregnancy was complicated by intrauterine growth restriction, oligohydramnios, and a small placenta. He was in the neonatal intensive care unit for 68 days post-delivery for complications including pulmonary hypertension, ventricular septal defect, failure to thrive, and multiple urinary tract infections. He had a gastrostomy-button placed at 13 months of age and continues to use a gastrostomy tube to supplement table feeds. He experienced his first seizure at 14 months of age. The seizures became well-controlled on Trileptal, he has not experienced a seizure since he was 5 years old, and was subsequently weaned off of anticonvulsants at 7 years of age. The proband was developmentally delayed, sitting at 12 months, walking at 26 months, and speaking his first words at 3 years of age. At 15 years of age, he was performing at a 5th-7th

grade level in home school. Other findings include short stature, pectus excavatum, chronic sinusitis and bronchiectasis, and an abnormal rash present since childhood, with a reticulated pattern that was poikilodermatous in some areas and ichthyotic in other areas. Karyotype and chromosomal microarray analyses were normal. Clinical WES was performed at GeneDx.

## Family KD-4

The probands (KD-4–3 and KD-4–4) are siblings from a consanguineous Persian family. The male proband (KD-4–4) was born at full term and weighed 2.65 kg with a length of 32 cm. He was delayed in his early developmental milestones and first walked at 6 years of age. He was 20 years old at the time of last examination and his growth parameters were: weight of 36 kg, height of 166 cm, and head circumference of 52.5 cm. He was diagnosed with ASD and remains nonverbal with severe intellectual disability. Neurological exam showed hypotonia, tremor, ataxia, abnormal gait, kyphoscoliosis, and difficulty feeding and swallowing.

The affected sister (KD-4–3) was born at full term and weighed 2.97 kg with a length of 49 cm. She sat unsupported at about 9 months and walked at about 27 months but she could not speak yet. She was 3 years old at the time of last examination and her growth parameters were: weight of 16 kg, height of 111 cm, and head circumference of 48.5 cm. She was diagnosed with ASD and neurological exam showed abnormal gait. Other findings included vision impairment (astigmatism and hyperopia) and a cleft palate for which she had surgery in infancy. There were no specific findings on brain MRI and metabolic testing was normal. Karyotyping and chromosomal microarray analysis for both siblings were normal. WES was performed as previously described (*Makrythanasis et al., 2018*).

## Family KD-5

The proband (KD-5–3) is a 4 year old female from a consanguineous family of Middle Eastern ancestry. She presented with ASD, intellectual disability, developmental delay, absent speech, and facial dysmorphisms. She was born at 35 weeks of gestation weighing 1.8 kg. She sat at 1 year of age and shuffled until she was able to walk at 4 years of age. Brain MRI at 2 years of age showed evidence of periventricular leukomalacia. At 4 years of age her weight was 16 kg, her height was 100 cm, and her head circumference was 47 cm indicative of microcephaly. Her dysmorphic features include frontal bossing, prominent eyes with blue sclera, lax lower eyelids, down slanted palpebral fissures, orbital hypertelorism, maxillary hypoplasia, hypoplastic ala nasi, smooth philtrum, micrognathia, and low set ears. Other findings include atrial septal defect secundum type, abnormal gait, sleep problems, hyperextensible joints, bilateral clinodactyly of the 5th fingers, syndactyly 2/3 toes, high degree exotropia, hyperpigmented skin in the pubic region, and nevoid telangiectasia on the left side of the back. Her hearing and her reflexes were normal. Chromosomal microarray analysis was normal. WES was performed as previously described (*Monies et al., 2019*).

## Family KD-6

The proband (KD-6–3) is a 40 year old female of European ancestry. She presented with severe intellectual disability, developmental delay, focal epilepsy, tetraspastic gait, and ataxia. She exhibited behavioral difficulties, including aggression, self-injurious behavior, reluctant eye contact, and rigid rituals. At present the proband can only say a few words. The healthy parents described a protracted birth with total arrest. When the proband was eventually born with the help of a vacuum extractor, she was cyanotic, weighed 3.52 kg, and was 52 cm long. Her seizure disorder started on the first day of life. She exhibited febrile seizures as well as isolated tonic and generalized tonic clonic seizures. Additional findings include microcephaly, tented upper and full lower lip, anteverted nares, hallux valgus as well as hands with short fingers and large palms. Brain MRI showed gliosis of the parieto-central pyramidal tracts and mild atrophy of the parietal region.

WES was performed after negative gene panel diagnostics. Exome enrichment was carried out with the BGI Exome capture kit and the library was sequenced (100 bp paired-end) on a BGISEQ-500 at BGI. Primary and secondary bioinformatic processing of the raw data was performed using Varfeed (Limbus) and the variants were then annotated using Varvis (Limbus). A coverage of 10X was achieved for 95%, 94%, and 96% of targeted sequences for the proband, father, and mother, respectively. A total of 106,516,290, 100,890,598, and 178,119,532 reads were obtained, with an

average read depth of 73X, 65X, and 141X for the proband, father, and mother, respectively. Single nucleotide variants and copy number variants were evaluated and all potential protein-influencing variants were prioritized with regard to their minor allele frequency in gnomAD, pathogenicity, inheritance modes, and gene attributes such as function and expression. Following variant filtration, only the *KDM5A* variant (c.4048C > T) met all pathogenicity criteria.

### Family KD-7

The proband (KD-7–3) is a 13 year old female from a United States family of Hispanic ancestry. She presented with severe intellectual disability, lack of speech, self-abusive behaviors, microcephaly, and dysmorphic facies, including flat midface, proptosis, low set ears, broad forehead, broad nasal tip, high nasal bridge, and high palate. She was born at 40 weeks of gestation. Her developmental milestones were significantly delayed with sitting unassisted, crawling, and walking at 8–10 years of age. She is proportionately undergrown, has a history of independent ambulation with a reciprocating shuffling-type gait, but progressive joint contractures and muscle weakness have necessitated assisted ambulation. Additional findings include coarctation of the aorta, heart murmur, dextroscoliosis, Marfanoid habitus, progressive joint contractures at the knees and elbows, and recurrent infections. Chromosomal microarray revealed a 28.92 Mb region of loss of heterozygosity and a 50 kb microdeletion encompassing the *KDM5A* locus at 12p13.33 (chr12:458,096–507,789). Karyotyping and targeted sequencing of *FGFR2* and *SKI* were normal.

## Patient lymphoblastoid cell lines

Cells were cultured in RPMI-1640 with 2 mM L-Glutamine, 10% fetal bovine serum (FBS), 1% v/v penicillin-streptomycin solution, and 1% v/v Fungizone. For immunoblotting, cells were lysed in SDS lysis buffer and western blot analysis was performed as described below.

## CRISPR knock-in of the KD-4 splice site mutation in HEK293T cells

We used CRISPR/Cas9 to introduce the splice site mutation of family KD-4 (c.2541+1G > T) into the *KDM5A* locus in HEK293T cells. Prior to the experiment, cells were authenticated through STR profiling and were confirmed negative for mycoplasma contamination. The following primers were used for the single guide RNA (sgRNA): 5'-CACCGCCAAGCTCGGCAAGTAAAGG-3' and 5'-AAACCCTTTACTTGCCGAGCTTGGC-3'. The sgRNA oligos were annealed and ligated into pX330 (pU6-(Bbsl)_CBh-Cas9-T2A-mCherry) (*Chu et al., 2015*) (Addgene plasmid # 64324, http://www.addgene.org/64324, RRID:Addgene_64324). The donor sequence used was: 5'-AACTAGTTTAGATATACCATTCTGTGAAACCCAGAACAGTGGACCATGTCACAAAGGACATTACCCAAAAAAAGGATGTAATACTCCAACTTTACTTGCCGAGCTTGGCTGATGACACACGGAAGAC-3'. The asymmetric donor DNA was optimized for annealing by overlapping the Cas9 cut site with 36 bp on the PAM-distal side and with a 91 bp extension on the PAM-proximal side of the break (*Richardson et al., 2016*).

HEK293T cells were cultured in DMEM with 10% FBS, 5% v/v penicillin-streptomycin solution, and 5% v/v Fungizone. Cells were transfected with polyethylenimine (PEI) in OptiMEM, and sorted using fluorescence-activated cell sorting (FACS) for mCherry fluorescence, present in the plasmid with Cas9 and the sgRNA. Individual clones were collected and sequenced to identify the knock-in. A successful knock-in clone and two wild type control clones were kept in culture for subsequent analyses.

## ENU mutagenesis and mouse breeding

All animal care and use procedures were approved by the UTSW Institutional Animal Care and Use Committee (protocol number 2017–102300) and were compliant with US Government principles about the care and use of animals, Public Health Service Policy on Humane Care and Use of Laboratory Animals, Guide for the Care and Use of Laboratory Animals, and the Animal Welfare Act. Animal husbandry was performed in the UTSW animal facility, accredited by the Association for Assessment and Accreditation of Laboratory Animal Care International. Male C57BL/6J mice were mutagenized with ENU and breeding was performed as previously described (*Wang et al., 2015*; *Figure 1—figure supplement 1*). Whole exome sequencing, genotyping, and automated mapping were performed as previously described (*Wang et al., 2015*).

## Generation of *Kdm5a* constitutive knockout mice

CRISPR/Cas9-mediated targeting of the *Kdm5a* locus was used to generate the *Kdm5a* knockout mouse model (*Kdm5a*$^{-/-}$). The sequence of the sgRNA used was: 5'-ttaatacgactcactatagGGGA TACAACTTTGCCGAAG-3'. The knockout allele contained a single base pair insertion in exon 13 shown in the square brackets (5'-ATAC AACTTTGCCG[G]AAGCGGTGAACTTCT-3'), predicted to result in a frameshift after amino acid 581 of KDM5A and terminating after the inclusion of 8 aberrant amino acids. For genotyping, tail genomic DNA was amplified across the insertion site and sequenced. The PCR primers were: 5'-ACTTACAGGACTTAA CTTAGGAAGTGAGTACA-3' and 5'-AAACTATCTATTCCCTCAGGAGAGACAAACA-3'; the sequencing primer was: 5'-CAGAAGCAGA-GACAGGTGG-3'. Constitutive *Kdm5a*$^{-/-}$ mice were bred as *Kdm5a*$^{+/-}$ x *Kdm5a*$^{+/-}$ for all experiments.

## Western blot analysis and antibodies

To detect endogenous proteins, cortical tissue was isolated from male and female mice and immediately frozen in liquid nitrogen. Tissue was lysed in SDS lysis buffer (250 mM Tris pH 6.8, 4% SDS, 3.2% glycerol, 10 mM NEM, 1 mM PMSF) supplemented with 2 mM OPT, and cOmplete mini EDTA-free protease inhibitor and phosSTOP phosphatase inhibitor tablets (Roche). Samples were boiled for 10 min (min), then centrifuged at 14,000 rpm for 15 min, and the supernatant was collected. Protein concentrations were determined using the DC protein assay (Bio-Rad). Samples were diluted in sample buffer and 50 µg of protein was loaded per lane, with β-mercaptoethanol and bromophenol blue, onto either 6% (for KDM5A) or 8% polyacrylamide gels. Gels were run and protein was transferred to polyvinylidene difluoride (PVDF) membranes (Millipore) for 2 hr (hr) on ice. Membranes were blocked in 5% milk in TBS-T (20 mM Tris pH 7.5, 150 mM NaCl, 0.1% Tween-20) for 1 hr at room temperature, and incubated with primary antibody overnight at 4°C. Membranes were washed in 5% TBS-T followed by a 1 hr incubation in secondary antibody (1:10,000) at room temperature using donkey anti-rabbit (Jackson ImmunoResearch, 711-035-152) or peroxidase AffiniPure donkey anti-mouse (Jackson ImmunoResearch, 715-035-150). Following antibody incubation, signal was detected with enhanced chemiluminescence (ECL; SuperSignal West Pico chemiluminescent substrate, Thermo Fisher Scientific). Primary antibodies used were: KDM5A (Abcam, ab70892, 1:1,000), vinculin (Cell Signaling, 4650S, 1:1,000), β-actin (Abcam, ab6276, 1:1,000).

## Golgi-Cox staining

Mice at 14–16 weeks of age were euthanized and brains were removed. Golgi-Cox staining was performed as described in *Zaqout and Kaindl, 2016*. The brains were cut into two hemispheres and impregnated with Golgi-Cox solution (1% $K_2Cr_2O_7$, 1% $HgCl_2$, 0.8% $K_2CrO_4$) at room temperature for 7 days in the dark. Subsequently, they were protected with tissue protectant solution (30% sucrose, 1% PVP40, 30% ethylene glycol in 0.05 M phosphate buffer pH 7.2) at 4°C for 24 hr after which the solution was replaced with a fresh one and kept at 4°C for 5 days. Brains were embedded in OCT medium. Frozen sagittal sections (150 µm thickness) were prepared using a microtome (Leica) and loaded on 3% gelatin-coated glass slides. The slides were dried for 5 days at room temperature, dehydrated, and developed as described in Zaqout et al. 2016 (*Zaqout and Kaindl, 2016*), and mounted with Cytoseal (Thermo Fisher Scientific). Brightfield images were acquired using Nikon Eclipse80i with 60X objective (zoom X1.4) at 0.5 µm steps in z axis. Dendritic and spine analyses were performed using Neurolucida 360 software (MBF Bioscience) at the UTSW Whole Brain Microscopy Facility. For Sholl analysis, images were processed and analyzed in Fiji (ImageJ v2.1.0/1.53 c) (*Schindelin et al., 2012*).

## Behavioral assays

All mice were age- and sex-matched littermate progeny of heterozygous *Kdm5a* mutant crosses. An experimenter blind to genotypes performed all behavioral tests. Ambient lighting was maintained at 60 lux. Mice were tested at 5–9 weeks of age except where indicated, and tests were performed in the order in which they appear below.

### Ultrasonic vocalization

USVs were recorded at P4 for both male and female pups isolated from their mothers during the daylight period of the light/dark cycle. Dams and their litters were acclimated to the test room for

30 min. Each pup was removed from the cage containing its mother and littermates and placed in a clean plastic container in a wooden sound-attenuating recording chamber. Each pup was first acclimated to the recording chamber for 30 s (sec) then recorded for 10 min. Recordings were acquired using an UltraSoundGate CM16/CMPA condenser microphone (Avisoft Bioacoustics) positioned at a fixed height of 8 cm above the pups, and were amplified and digitized (sampled at 16 bits, 250 kHz) using UltraSoundGate 416 hr 1.1 hardware and Avisoft-RECORDER software (Avisoft Bioacoustics). The data were transferred to Avisoft-SASLab Pro (version 5.2) to analyze spectrograms of vocalizations with settings of 0% overlapping FlatTop windows, 100% frame size, and 256 points fast Fourier transform (FFT) length. The following measures were recorded and analyzed for each group: number of USV calls, mean duration of USV calls, mean amplitude, mean peak frequency, maximum peak frequency, mean interval between USV calls, and the latency to call.

### Nest building
Test mice, at 4 weeks of age, were single-housed in their home cages right before the dark phase, and one nestlet was placed in each cage on day 1. At the beginning of the light phase on the next day (12 hr time point), the quality of the nest was examined, and the nest was photographed. The nest quality was subsequently assessed at 24 hr, 36 hr, 48 hr, and 72 hr time points. Nest quality was scored on a scale of 0 to 3 at each time point (0 being no nest construction, three being a well-structured nest) (*Figure 2—figure supplement 2B*).

### Forepaw wringing
We used a modified version of the tail suspension test to quantify forepaw wringing and/or clasping. Test mice were acclimated to the test room for 30 min. Each mouse was suspended by the tail and videotaped for 3 min. Time spent forepaw wringing and/or clasping was measured, and data was plotted as percent of total time spent forepaw wringing and/or clasping.

### Grooming
Mice were placed into a new standard cage, without nestlets, food, or water, acclimated for 10 min, then videotaped for another 10 min. The amount of time spent grooming was recorded continuously to calculate the total time spent grooming. Grooming was considered self-grooming of any part of the body (including the face, head, ears, body, or tail).

### Open field test
Mice were acclimated to the test room for 30 min. The open field apparatus consisted of a white square box ($45.7 \times 45.7 \times 30.5$ cm) with photo beams to record horizontal and vertical movements of the mouse. Mice were individually placed in the center of the box. Activity was quantified over a 15 min period by EthoVision XT 13 software. Total distance traveled, time spent in the periphery of the box, and time spent in the non-periphery of the box were recorded.

### Direct social interaction
Test mice were single-housed in a standard housing cage with food and water for one night. The next day, mice were acclimated to the test room for 30 min and then a conspecific interactor mouse of the same sex and age was introduced into the housing cage. The activity of the mice was videotaped by EthoVision XT 13 software for 10 min and the total time the test mouse spent in direct social contact with the conspecific interactor mouse was manually recorded by the observer. Direct social contact included nose-nose, anogenital, or body sniffing.

### Morris water maze (MWM) test
Mice were acclimated to the test room for 30 min. MWM was carried out in a 144 cm diameter pool filled with room temperature water (~24°C) and made opaque with harmless white paint. A platform was submerged in the pool, not visible to mice while swimming. During the training phase, each mouse was given eight trials per day, in blocks of 4 trials, for four consecutive days, each time placed in the pool at a different starting point. The time taken to locate the hidden escape platform (latency) and the distance traveled were measured. For the probe test, the platform was removed and each mouse was given 1 min to search the pool. The amount of time each mouse spent in each

quadrant was recorded. The mice were tracked during the training trials and probe test using Etho-Vision XT 13 software.

## RNA sequencing and data analysis

RNA was extracted from hippocampal tissue of 10 animals at 6–8 weeks of age (5 WT and 5 Kdm5a$^{-/-}$) using Trizol reagent (Invitrogen) and RNeasy Mini Kit (Qiagen) and DNA was removed using RNase-free DNase kit (Qiagen). Samples were run on the Agilent Tapestation 4200 to assess quality and those with a RIN score of ≥8 were used for sequencing. Concentrations were measured on a Qubit fluorometer and 1 µg of total DNase-treated RNA was used for library preparation using the TruSeq Stranded mRNA Library Prep Kit from Illumina. Poly-A RNA was purified, fragmented, and strand specific samples were amplified and purified with AmpureXP beads. Samples were sequenced on the Illumina NextSeq 500 using V2.5 reagents. Raw data from the machine were de-multiplexed and converted to FASTQ files using bcl2fastq (v2.17, Illumina). The FASTQ files were checked for quality using FastQC (v0.11.2) and FastQ_Screen (v0.4.4). Quality control metrics are presented in *Figure 5—figure supplement 1*. Reads were mapped to mm10 reference Mouse genome using STAR (v2.5.3a) (*Dobin et al., 2013*). Read counts were then generated using feature-Counts (Subread v1.4.6) (*Liao et al., 2014*). TMM normalization and differential expression analysis were performed using edgeR (v3.18.1) (*Robinson et al., 2010*). Hierarchical clustering of differen-tially expressed genes was performed using hclust (R v3.4.0 'You Stupid Darkness') and heatmaps were generated using heatmap2 (gplots v3.0.1). Principal component analysis was performed using ClustVis (*Metsalu and Vilo, 2015*). GO analysis was performed using DAVID 6.8 (*Huang et al., 2009a*; *Huang et al., 2009b*) with the mouse genome as the background and redundant gene ontol-ogy terms were merged using Revigo (*Supek et al., 2011*). The list of known ASD genes was obtained from the SFARI Gene database (*Banerjee-Basu and Packer, 2010*), and the list of neurode-velopmental genes was obtained from the BrainSpan Atlas of the Developing Human Brain (*Miller et al., 2014*).

## Quantitative RT-PCR

RNA was extracted from hippocampal tissue of 10 animals at 6–8 weeks of age (5 WT and 5 Kdm5a$^{-/-}$) and 1 µg of RNA was reverse transcribed into cDNA using Superscript III First Strand Syn-thesis System for RT-PCR (Invitrogen). Real-time quantitative PCR was performed using PowerUp SYBR Green (2X) master mix (Applied Biosystems) and the following primers:

*Efcab6* forward: 5'-CTGGAGCAGTGAGGGTCAAC-3', and reverse: 5'-ATGGTCCCCGTG TCCCTAAG-3';
*Ptgds*, forward: 5'-GCTCCTTCTGCCCAGTTTTC-3', and reverse: 5'-CCCCAGGAACTTGTC TTGTTGA-3';
*Shh*, forward: 5'-ACTGGGTCTACTATGAATCC-3', and reverse: 5'-GTAAGTCCTTCACCAGC TTG-3';
*Ccnd1*, forward: 5'-TTCCCTTGACTGCCGAGAAG-3', and reverse: 5'-AAATCGTGGGGAGTCA TGGC-3'.

## Statistical analysis

All statistical analyses were performed using GraphPad Prism (http://www.graphpad.com/scientific-software/prism, RRID:SCR_002798). All analyses were performed using at least three different ani-mals of each genotype or three replicates for cell lines. Data are represented as mean ± SEM and two-tailed *P* values are reported. GraphPad Prism was used to perform either one-way ANOVA (fac-tor: genotype), two-way ANOVA (factors: genotype and age or time), or a mixed-effects model (restricted maximum likelihood; REML) (factors: genotype and age or time) with Tukey's multiple comparisons test, or ratio paired or unpaired t test where applicable. All figures were prepared using Adobe Illustrator (http://www.adobe.com/products/illustrator, RRID:SCR_010279).

## Electronic resources

Allen Brain Atlas Mouse Brain: https://mouse.brain-map.org
BrainSpan: https://www.brainspan.org
FastQC: https://www.bioinformatics.babraham.ac.uk/projects/fastqc/

Fiji: https://fiji.sc/ gnomAD Browser: http://gnomad.broadinstitute.org
GeneMatcher: https://genematcher.org/
GTEx Portal: https://www.gtexportal.org/home
MGI: http://www.informatics.jax.org/
OMIM: http://www.omim.org
SFARI Gene: https://gene.sfari.org/database/human-gene/
UCSC Genome Browser: http://genome.ucsc.edu

### Contact for reagent and resource sharing

Further information and requests for resources and reagents should be directed to and will be fulfilled by the Lead Contact, Maria H. Chahrour (maria.chahrour@utsouthwestern.edu).

## Acknowledgements

We are grateful to the families for their participation in our study. We thank Daniel A Schmitz and Carlos A Pinzon-Arteaga for technical help and advice in designing the CRISPR knock-in experiment. We thank Chelsea Burroughs for assistance in preparing the Figures. The CAUSES Study investigators include Shelin Adam, Christele Du Souich, Alison Elliott, Jan Friedman, Anna Lehman, Jill Mwenifumbo, Tanya Nelson, and Clara Van Karnebeek. The CAUSES Study was funded by Mining for Miracles, British Columbia Children's Hospital Foundation, and Genome British Columbia. This work was supported by the University of Texas Southwestern Medical Center and grants from the Welch Foundation and the Walter and Lillian Cantor Foundation to MHC.

## Additional information

### Funding

| Funder | Grant reference number | Author |
| --- | --- | --- |
| Welch Foundation | I-1946-20180324 | Maria H Chahrour |
| Walter and Lillian Cantor Foundation | | Maria H Chahrour |
| University of Texas Southwestern Medical Center | | Maria H Chahrour |

The funders had no role in study design, data collection and interpretation, or the decision to submit the work for publication.

### Author contributions

Lauretta El Hayek, Conceptualization, Data curation, Software, Formal analysis, Validation, Investigation, Visualization, Methodology, Writing - original draft, Writing - review and editing; Islam Oguz Tuncay, Software, Formal analysis, Validation, Investigation, Writing - review and editing; Nadine Nijem, Software, Formal analysis, Investigation, Methodology, Writing - review and editing; Jamie Russell, Sara Ludwig, Amanda Gerard, Anja Heinze, Pia Zacher, Hessa S Alsaif, Aboulfazl Rad, Kazem Hassanpour, Mohammad Reza Abbaszadegan, Camerun Washington, Barbara R DuPont, Raymond J Louie, CAUSES Study, Madeline Couse, Maha Faden, R Curtis Rogers, Rami Abou Jamra, Ellen R Elias, Reza Maroofian, Anna Lehman, Resources, Investigation, Writing - review and editing; Kiran Kaur, Formal analysis, Investigation, Methodology, Writing - review and editing; Xiaohong Li, Priscilla Anderton, Miao Tang, Resources, Investigation, Methodology, Writing - review and editing; Henry Houlden, Resources, Writing - review and editing; Bruce Beutler, Conceptualization, Resources, Supervision, Validation, Writing - review and editing; Maria H Chahrour, Conceptualization, Resources, Data curation, Formal analysis, Supervision, Funding acquisition, Validation, Investigation, Visualization, Methodology, Writing - original draft, Project administration, Writing - review and editing

## Author ORCIDs

Lauretta El Hayek 
Aboulfazl Rad 
Maria H Chahrour 

## Ethics

Human subjects: The study was approved by the University of Texas Southwestern Medical Center Institutional Review Board (protocol number STU 032015-014). Written informed consent was obtained from all study participants.

Animal experimentation: All animal care and use procedures were approved by the University of Texas Southwestern Medical Center Institutional Animal Care and Use Committee (protocol number 2017-102300).

## Decision letter and Author response

Decision letter https://doi.org/10.7554/eLife.56883.sa1
Author response https://doi.org/10.7554/eLife.56883.sa2

# Additional files

## Supplementary files

- Transparent reporting form

## Data availability

All data are available in the main text or the supplementary materials. The gene expression data reported in this study are available in the NCBI Gene Expression Omnibus (GEO) repository with accession GSE147435.

The following dataset was generated:

| Author(s) | Year | Dataset title | Dataset URL | Database and Identifier |
|---|---|---|---|---|
| El Hayek L, Chahrour MH | 2020 | Impaired transcriptional profile in the hippocampus following loss of *Kdm5a* | https://www.ncbi.nlm.nih.gov/geo/query/acc.cgi?acc=GSE147435 | NCBI Gene Expression Omnibus, GSE147435 |

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
