## [Decision Letter]

**Acceptance summary:**

El Hayek and colleagues pursued an elegant forward genetics screening approach to identify *KDM5A*, which encodes a histone lysine demethylase, as a possible autism-spectrum-disorder-related gene. The study provides interesting new information on the genetic background of neurodevelopmental disorders and represents a valuable contribution to the field of neuropsychiatric genetics.

**Decision letter after peer review:**

Thank you for submitting your article "*KDM5A* mutations identified in autism spectrum disorder using forward genetics" for consideration by *eLife*. Your article has been reviewed by two peer reviewers, and the evaluation has been overseen by a Reviewing Editor and Huda Zoghbi as the Senior Editor. The following individual involved in review of your submission has agreed to reveal their identity: Hannelore Ehrenreich (Reviewer #1).

The reviewers have discussed the reviews with one another and the Reviewing Editor has drafted this decision to help you prepare a revised submission.

Summary:

El Hayek employed a forward genetics approach in mice to identify *Kdm5a* as a candidate ASD gene. Behavioral analysis of carriers of the mutation revealed altered ultrasonic vocalization (USV) along with impaired nest-building behavior, as known behavioral features of ASD mouse models. Constitutive genetic ablation of *Kdm5a* (*Kdm5a*^-/-^ or KO) confirmed the physiological relevance of the gene with respect to vocalization. The observed increased clasping in a modified version of the tail suspension test is interpreted as a measure of stereotypic behavior. Hippocampal transcripts of WT and KO mice were assessed and compared to each other, resulting in a total of 450 dysregulated genes in KO mice, of which 191 were up- and 259 downregulated. The latter were mostly associated with neurogenesis and cell proliferation, while upregulated genes largely reflected genes related to RNA splicing and neurological processes. The findings of this manuscript point to *Kdm5a* as a possible candidate gene for neurodevelopmental disorders.

All reviewers agree that your identification of *KDM5A* as a possible ASD-related gene has been elegantly conducted and that your study provides interesting new information on the genetic background of neurodevelopmental disorders. The reviewers support ultimate publication of your study in *eLife*, but a series of issues need to be addressed before a final decision can be made.

Essential revisions:

Behavioral Testing

1) A general issue is that only very few ASD-specific behavioral tests were conducted; social behavior was not tested systematically, and neither was cognitive performance. In view of these limitations, the current dataset indicates a role of *KDM5A* variants in a general and as yet undefined neurodevelopmental disorder rather than ASD specifically. This notion is not contradicted by the fact that pathogenic *KDM5A* variants are detected in patients with ASD since other genetic variations or even non-genetic causes may have contributed to the clinical picture, which resembles a neurodevelopmental syndrome with autistic features.

2) Throughout all mouse studies, inconsistent and often problematically low numbers of animals of different genotypes (WT vs. Het vs. KO) for behavioral and particularly statistical analysis are used (e.g. Figure 2—figure supplements 1 and 2). Here, additional experiments are necessary to bolster the statistics.

3) In the linkage analysis for maximum peak frequency of USV and nesting score (Figure 1), the authors used only 3 KO (Mut) mice, while using 17 WT and 24 Het mice. It is unclear whether a low number of KO mice (n=3) is sufficiently representative in this approach or may possibly cause an analytical misrepresentation. This needs to be rectified.

The issues mentioned above (1-3) need to be addressed to bolster the notion of *KDM5A* variants being involved in an as yet undefined neurodevelopmental disorder.

4) If a specific link to ASD is to be made, more experiments are necessary:

– The tail suspension test does not apply to the measurement of repetitive and/or stereotypic phenotypes. Alternative assays for repetitive and/or stereotypic phenotypes (e.g. marble burying tests, standardized observation of grooming, circling behavior) should be performed.

– The clasping assay reads out multiple features, including central paresis, and its usefulness and specificity as a reliable measure of stereotypic behavior in ASD mouse models is questionable. The authors refer to Chao et al., 2010, in this regard, but the respective reference in this article is not obvious. This needs to be clarified.

– The authors only performed a very limited set of assays that assess ASD-related behavior. A more detailed analysis of ASD-related behavioral features is required, most importantly social interaction/preference (e.g. in tripartite chamber).

– Given that *KDM5A* function is thought to be associated with memory formation and retrieval, tests for cognitive disabilities (spatial learning, working memory, object memory, social memory) should be performed – for 'differential diagnosis'.

The experiments summarized under (4) would not be necessary if the thrust of the paper were changed from the ASD focus toward a more general neurodevelopmental disorder.

Statistics

5) It has to specified in every instance – in the text and legends – which statistical test was used to analyze the corresponding datasets.

6) A systematic justification of the adequacy of statistical tests used is required in every case. Some statistical tests have to be re-interpreted, and different statistical tests are required in several instances to address the issues at hand. In Figure 2—figure supplement 1, for instance: In order to show that there was no significant difference in body weight across time and genotypes, a mixed ANOVA has to be applied. The currently described 1-way ANOVA seems inappropriate in this context – and it is unclear on which data and how this ANOVA was performed. Similarly, in the conclusion that "Male and female KO mice do not show any significant change in body length compared to WT or Het littermates" is not possible using just ANOVA, which is an omnibus test. For the current conclusion, a post-hoc test would be necessary. Alternatively, the conclusion should be adjusted to "do not show any significant differences in body length between genotypes". For an ANOVA to be justified (instead of a non-parametric alternative), it needs to be tested whether assumptions are held.

Genetics

7) Whole-exome sequencing and microarray data details from patients are required.

*Kdm5a* KO Studies8) The characterization of the *KDM5A* mutants is still rather superficial. It is said that the chromatin remodeler *KDM5A* is leading to a neurodevelopmental disturbance, but no data on protein expression of *KDM5A* and other neuronal proteins or on brain development and morphology are provided. Moreover, the expression of the other genes of the KDM family (which are also leading to neurodevelopmental disorders when mutated) are not analysed. This should be rectified by adding corresponding additional data on the KO.

9) The RNA profiling data are simply listed and the putative targets are described. The validation of some of the targets is, however, missing so that in the end this part of the study is not more than a starting point for the identification of a *KDM5A* related pathomechanism. At the least, the misregulation of a select small subset of genes needs to be validated (e.g. by Western blotting).

---

## [Author Response]

Essential revisions:Behavioral Testing1) A general issue is that only very few ASD-specific behavioral tests were conducted; social behavior was not tested systematically, and neither was cognitive performance. In view of these limitations, the current dataset indicates a role of KDM5A variants in a general and as yet undefined neurodevelopmental disorder rather than ASD specifically. This notion is not contradicted by the fact that pathogenic KDM5A variants are detected in patients with ASD since other genetic variations or even non-genetic causes may have contributed to the clinical picture, which resembles a neurodevelopmental syndrome with autistic features.

The pathogenic mutations that we identified in the reported patients are the only pathogenic variants that segregated with phenotype in these families. We now clarify this in the Results section and explicitly state that apart from the *KDM5A* variants that we identified, there were no variants in known disease-causing genes that accounted for the clinical phenotype of the nine individuals. Furthermore, we identified seven independent alleles in nine patients with ASD (Table 1 and Figure 4). Importantly, all the single nucleotide variants that we identified in the patients were absent from gnomAD, a database of 125,748 whole exome and 15,708 whole genome sequences from unrelated individuals, further strengthening the evidence for the pathogenicity of these *KDM5A* variants. In fact, according to gnomAD data, the probability of loss-of-function intolerance (pLI) for *KDM5A* is 1.00, and there is strong selection against predicted loss-of-function variation in *KDM5A* (LOEUF score=0.16) (Definition of LOEUF from gnomAD: LOEUF: loss-of-function observed/expected upper bound fraction. It is a conservative estimate of the observed/expected ratio, based on the upper bound of a Poisson-derived confidence interval around the ratio. Low LOEUF scores indicate strong selection against predicted loss-of-function variation in a given gene, while high LOEUF scores suggest a relatively higher tolerance to inactivation).

In addition, for 5 out of the 7 identified *KDM5A* variants, we show a direct effect of the variant on KDM5A protein levels (Figure 4, Figure 4—figure supplement 5, and Figure 4—figure supplement 6), demonstrating that these variants are damaging. According to current clinical genetics guidelines for the interpretation of sequence variants (Richards et al., 2015), our data constitute a preponderance of evidence in favor of the conclusion that the identified *KDM5A* variants are pathogenic and causative.

Furthermore, as suggested by the reviewer, and detailed in our subsequent responses below, we now present data demonstrating that the *Kdm5a* knockout mice have deficits in sociability and cognitive function (Figure 2 and Figure 2—figure supplement 4).

2) Throughout all mouse studies, inconsistent and often problematically low numbers of animals of different genotypes (WT vs. Het vs. KO) for behavioral and particularly statistical analysis are used (e.g. Figure 2—figure supplements 1 and 2). Here, additional experiments are necessary to bolster the statistics.

We bolstered the number of mice per experiment and added the new data to the revised manuscript.

3) In the linkage analysis for maximum peak frequency of USV and nesting score (Figure 1), the authors used only 3 KO (Mut) mice, while using 17 WT and 24 Het mice. It is unclear whether a low number of KO mice (n=3) is sufficiently representative in this approach or may possibly cause an analytical misrepresentation. This needs to be rectified.

Since the initial screen was a forward genetics screen (Figure 1 and Figure 1—figure supplement 1), we had no control over the number of animals tested per genotype. The behavioral screening is performed in pedigrees and is blinded to genotype, and the genotyping, mapping, and linkage are performed subsequently. Nonetheless, the data was highly statistically significant and linkage was detected (*P*=2.9 X 10^-5^ for the USV phenotype and *P*=7.0 X 10^-5^ for the nesting phenotype) despite the n=3 mutant mice (Figures 1D and E). Furthermore, in the validation experiment, where we directly and specifically targeted the *Kdm5a* locus, we confirmed the USV phenotype by analyzing additional mice (n=9 knockout mice; Figure 2 and Figure 2—figure supplement 4). The mutant mice in Figure 1 are part of the ENU mutagenesis screen, while the *Kdm5a* knockout mice analyzed in the rest of the experiments were generated via targeted disruption of the *Kdm5a* locus, thus our findings in the knockout mice represent independent validation of the behavioral phenotypes identified in the mutagenized mice.

The issues mentioned above (1-3) need to be addressed to bolster the notion of KDM5A variants being involved in an as yet undefined neurodevelopmental disorder.4) If a specific link to ASD is to be made, more experiments are necessary:– The tail suspension test does not apply to the measurement of repetitive and/or stereotypic phenotypes. Alternative assays for repetitive and/or stereotypic phenotypes (e.g. marble burying tests, standardized observation of grooming, circling behavior) should be performed.

As suggested by the reviewer, we measured self-grooming and found that the *Kdm5a* knockout mice spent twice as much time self-grooming compared to control littermates. We included this new data in Figure 2E.

– The clasping assay reads out multiple features, including central paresis, and its usefulness and specificity as a reliable measure of stereotypic behavior in ASD mouse models is questionable. The authors refer to Chao et al., 2010, in this regard, but the respective reference in this article is not obvious. This needs to be clarified.

During our routine handling of the *Kdm5a* knockout (KO) mice, we noticed a distinct repetitive forepaw wringing and clasping behavior. We quantified this behavior and found that the KO mice spend significantly more time wringing and clasping their forepaws (60% of the time) compared to their control littermates (4-5% of the time) (Figure 2D). This repetitive wringing and clasping behavior has been observed in mouse models of several neurodevelopmental disorders. In Chao et al., 2010, the authors characterize this behavior in a mouse model generated to study Rett syndrome (Figure 1D in Chao et al., 2010). We reference the study as an example of paw wringing and clasping behavior in mouse models of ASD and related neurodevelopmental disorders.

– The authors only performed a very limited set of assays that assess ASD-related behavior. A more detailed analysis of ASD-related behavioral features is required, most importantly social interaction/preference (e.g. in tripartite chamber).

Of the behavioral assays we performed, analyses of ultrasonic vocalizations (USVs) and nest building ability are often used to uncover social deficits in mice. USV quantification is a well-documented tool to assess social communication, social interest, and motivation in mice (Branchi et al., 2001, Portfors, 2007, Vogel et al., 2019). Abnormalities in these socially evoked USVs are typically seen in mouse models of autism and related neurodevelopmental disorders and are indicators of social behavior deficits and abnormal brain development (Crawley, 2007, Scattoni et al., 2009). Another indicator of sociability in mice is the ability to build a communal nest and their inability to do so demonstrates social deficits (Crawley, 2007). Assessment of USVs and nest building ability identified defects in these behaviors in the *Kdm5a* knockout mice compared to littermate controls (Figures 1 and 2, and Figure 2—figure supplement 4).

Furthermore, as suggested by the reviewer, we assessed sociability of the *Kdm5a* knockout mice through a direct social interaction test, which allows for detailed assessment of intrinsic mouse interactions and is commonly used to assess social interactions in ASD mouse models (Crawley, 2012, Chang et al., 2017). We found that the knockout mice spend significantly less time interacting with a novel partner compared to their control littermates (Figure 2G). We included this new data in Figure 2.

– Given that KDM5A function is thought to be associated with memory formation and retrieval, tests for cognitive disabilities (spatial learning, working memory, object memory, social memory) should be performed – for 'differential diagnosis'.

To assess cognitive function in *Kdm5a* knockout mice, we performed the Morris water maze test. We found that the knockout mice failed to learn the location of the hidden platform during the training phase compared to their control littermates (Figure 2H). In addition, the knockout mice had a significant deficit in memory in the probe test compared to their control littermates (Figure 2I). We included this new data in Figure 2.

The experiments summarized under (4) would not be necessary if the thrust of the paper were changed from the ASD focus toward a more general neurodevelopmental disorder.Statistics5) It has to specified in every instance – in the text and legends – which statistical test was used to analyze the corresponding datasets.

We listed all the specific statistical tests used in the figure legends and the Materials and methods.

6) A systematic justification of the adequacy of statistical tests used is required in every case. Some statistical tests have to be re-interpreted, and different statistical tests are required in several instances to address the issues at hand. In Figure 2—figure supplement 1, for instance: In order to show that there was no significant difference in body weight across time and genotypes, a mixed ANOVA has to be applied. The currently described 1-way ANOVA seems inappropriate in this context – and it is unclear on which data and how this ANOVA was performed. Similarly, in the conclusion that "Male and female KO mice do not show any significant change in body length compared to WT or Het littermates" is not possible using just ANOVA, which is an omnibus test. For the current conclusion, a post-hoc test would be necessary. Alternatively, the conclusion should be adjusted to "do not show any significant differences in body length between genotypes". For an ANOVA to be justified (instead of a non-parametric alternative), it needs to be tested whether assumptions are held.

We agree with the reviewer and thank them for pointing this out. As suggested by the reviewer, we conducted the appropriate mixed-effects model test (restricted maximum likelihood; REML) for repeated measures followed by Tukey's multiple comparisons test for the body weight data in Figure 2—figure supplement 3A. We also performed a post-hoc test for the one-way ANOVA for the data in Figure 2—figure supplement 3B and Figure 2—figure supplement 3C.

Genetics7) Whole-exome sequencing and microarray data details from patients are required.

We added all the details for the whole exome sequencing and microarray data in the Materials and methods under “Subjects and specimens” and “Clinical presentation”.

Kdm5a KO Studies8) The characterization of the KDM5A mutants is still rather superficial. It is said that the chromatin remodeler KDM5A is leading to a neurodevelopmental disturbance, but no data on protein expression of KDM5A and other neuronal proteins or on brain development and morphology are provided. Moreover, the expression of the other genes of the KDM family (which are also leading to neurodevelopmental disorders when mutated) are not analysed. This should be rectified by adding corresponding additional data on the KO.

We thank the reviewer for this comment. We had presented data on the loss of KDM5A protein from *Kdm5a* knockout tissue (Figure 2B). As suggested by the reviewer, we analyzed the neuronal phenotype of *Kdm5a* knockout mice using Golgi-Cox staining and we included this new data in Figure 3. We found that cortical layer II/III neurons from the knockout mice had significant reduction in dendritic complexity, length, and spine density, compared to neurons from WT littermates.

Furthermore, we found that many genes that function in and regulate neurological processes, including neurogenesis, are dysregulated upon loss of KDM5A in the knockout mice. Among the dysregulated genes we identified *Shh*, which plays a role in early nervous system development and regulates adult hippocampal neurogenesis and neural progenitor cell proliferation and differentiation. Dysregulation of *Shh* in adult mice has been shown to disrupt neurogenesis. We validated the upregulation of *Shh*, and its downstream target *Ccnd1*, in the *Kdm5a* knockout with qRT-PCR and included this new data in Figure 5C. Thus, the neuronal (Figure 3) and behavioral (Figure 2 and Figure 2—figure supplement 4) phenotypes and the transcriptional changes (Figure 5) observed in the *Kdm5a* knockout mice indicate that KDM5A plays an important role in neuronal development and function.

In addition, as suggested by the reviewer, we now provide data on the expression of the other KDM5 family members (*Kdm5b*, *Kdm5c*, and *Kdm5d*), demonstrating that their levels do not change in the *Kdm5a* knockout, suggesting no compensation from other KDM5 family members following loss of *Kdm5a* (Figure 5—figure supplement 9).

9) The RNA profiling data are simply listed and the putative targets are described. The validation of some of the targets is, however, missing so that in the end this part of the study is not more than a starting point for the identification of a KDM5A related pathomechanism. At the least, the misregulation of a select small subset of genes needs to be validated (e.g. by Western blotting).

We validated the dysregulation of four genes by quantitative RT-PCR and included this new data in Figure 5C. We found that *Shh* and its downstream target *Ccnd1* are upregulated upon loss of *Kdm5a*. Given the role of *Shh* in neuronal development, the data suggests a potential mechanism by which loss of *Kdm5a* gives rise to the observed neurobehavioral phenotypes.